# Nucleus reuniens transiently synchronizes memory networks at beta frequencies

Maanasa Jayachandran[1,4], Tatiana D. Viena [1,4], Andy Garcia [1], Abdiel Vasallo Veliz [1], Sofia Leyva [1], Valentina Roldan [1], Robert P. Vertes[2] & Timothy A. Allen [1,3] ✉

Episodic memory-based decision-making requires top-down medial prefrontal cortex and hippocampal interactions. This integrated prefrontal-hippocampal memory state is thought to be organized by synchronized network oscillations and mediated by connectivity with the thalamic nucleus reuniens (RE). Whether and how the RE synchronizes prefrontal-hippocampal networks in memory, however, remains unknown. Here, we recorded local field potentials from the prefrontal-RE-hippocampal network while rats engaged in a non-spatial sequence memory task, thereby isolating memory-related activity from running-related oscillations. We found that synchronous prefrontal-hippocampal beta bursts (15–30 Hz) dominated during memory trials, whereas synchronous theta activity (6–12 Hz) dominated during non-memory–related running. Moreover, RE beta activity appeared first, followed by prefrontal and hippocampal synchronized beta, suggesting that prefrontal-hippocampal beta could be driven by the RE. To test whether the RE is capable of driving prefrontal-hippocampal beta synchrony, we used an optogenetic approach (retroAAV-ChR2). RE activation induced prefrontal-hippocampal beta coherence and reduced theta coherence, matching the observed memory-driven network state in the sequence task. These findings are the first to demonstrate that the RE contributes to memory by driving transient synchronized beta in the prefrontal-hippocampal system, thereby facilitating interactions that underlie memory-based decision-making.

Episodic memory is adaptive in that it influences future behaviors, and depends on interactions between the medial prefrontal cortex and hippocampus[1–3]. A general axiom is that the prefrontal cortex exercises control over memories represented with spatiotemporal contexts in the hippocampus[4,5]; that is, information relayed from the prefrontal cortex guides situation-specific acquisition and retrieval, while the hippocampus provides context-specific memory[1,6–8]. Higher-order thalamocortical connections through the nucleus reuniens (RE)

contribute to memory by mediating dynamic activity states in prefrontal-hippocampal circuits[3,6,9–16].

Anatomically, the hippocampus projects to the prefrontal cortex[17], but return projections are sparse[18]. Memory-related interactions rather depend on the RE, a region of the midline thalamus that provides bi-directional control over prefrontal-hippocampal loops[10–12,19,20]. Theoretically, the RE organizes oscillatory synchrony, thereby mediating prefrontal-hippocampal interactions[2,10]. In this

[1]Cognitive Neuroscience Program, Department of Psychology, Florida International University, Miami, FL 33199, USA. [2]Center for Complex Systems and Brain Sciences, Florida Atlantic University, Boca Raton, FL 33431, USA. [3]Department of Environmental & Occupational Health, Robert Stempel College of Public Health, Florida International University, Miami, FL 33199, USA. [4]These authors contributed equally: Maanasa Jayachandran, Tatiana D. Viena. ✉e-mail: tallen@fiu.edu

regard, RE phase-locks to hippocampal theta (6–12 Hz) during spatial alternations[6,12,21], imposes slow oscillations in the hippocampal CA1 region[22,23], couples with delta (1–4 Hz)[22,24,25], and organizes cortical gamma (30–90 Hz)[22,23]. The RE is also critical to memory acquisition and retrieval in a broad array of tasks related to episodic memory[3,21,26].

Theta oscillations are extensively studied in learning and memory[27–30]. Theta coordinates activity during waking behaviors and rapid eye movement sleep, and organizes synaptic integration windows across distal regions[27,31–33]. However, running behaviors alone have been shown to drive massive theta[34,35]. Therefore, as most neurophysiological studies of memory in rodents involve running, this may obscure other memory-related rhythms[28,34–36]. Growing evidence indicates a distinct role of beta (15–30 Hz) activity in memory networks[36–41]. Currently available data suggest that beta originates from central drivers and supports flexible memory via sensorimotor integration and working memory[42,43]. Beta may thus be a key mechanism for maintaining synaptic information states in memory by bridging delays between discontinuous events or actions[42]. Whether and how beta synchronizes in prefrontal-hippocampal networks during memory, and the involvement of the RE, remain unknown.

## Results

We first evaluated whether and when the prefrontal-RE-hippocampal network synchronizes during non-spatial sequence memory, focusing on beta and theta rhythms. We used a nonspatial odor-based sequence memory task to model episodic memory of the flow of events in an experience, separating memory-related activity from running/navigational activity[3,36]. The task requires rapid retrieval from long-term memory stores and depends on the prefrontal-RE-hippocampal circuitry[3].

We recorded local field potentials (LFP) from 2 groups of rats ($n = 9$) implanted with dual-site silicon probes (32-channel arrays) targeting prefrontal-hippocampal and RE-hippocampal location pairs to characterize rhythmic mechanisms throughout the prefrontal-RE-hippocampal network during memory. Electrodes were positioned in deep layers of the prelimbic cortex (Supplementary Fig. 1A$_i$), the RE (Supplementary Fig. 1B$_i$), and/or stratum lacunosum moleculare in dorsal CA1 (Supplementary Fig. 1A$_{ii}$, B$_{ii}$). These regions are directly interconnected through glutamatergic projections, although there are only sparse prefrontal projections to CA1[11,17,20]. The RE innervates both glutamatergic and GABAergic target cells in the prefrontal cortex and CA1 and can control the excitatory-inhibitory tone from a distance[10,44–55].

During recordings, rats demonstrated their memory by making in-sequence (nose-poke >1 s; InSeq) and out-of-sequence (nose-poke <1 s; OutSeq) decisions (Fig. 1A). Sequence memory was measured using an established sequence memory index (SMI)[3,36]. Sequence memory was evident throughout all recordings (experiment-wide: SMI = 0.28 ± 0.01; $t_{(26)} = 24.62$, $p = 1.53 \times 10^{-19}$; all individual subject G-tests<0.05)[3,36]. Recordings were performed after memory reached pre-surgical levels (experiment-wide: $t_{(74)} = 1.16$, $p = 0.25$). In the prefrontal-hippocampal and RE-hippocampal experiments, the non-mnemonic behaviors were stable throughout (Supplementary Fig 2A).

### Prefrontal-hippocampal beta in memory

To test how prefrontal-hippocampal LFP modes relate to overt behavior states[34] we divided our analyses into 3 conditions: (1) memory (correct odor trials regardless of sequential context), (2) maze running (running between sequences), and (3) maze stationary (stationary between sequences not including periods around the nose port) (Fig. 1C, Supplementary Fig 3A).

We focused on low frequencies (1–50 Hz), including a priori beta (15–30 Hz) and theta (6–12 Hz) bands[36]. Figure 1D shows representative spectrograms. Overall, prefrontal cortex and hippocampal sites

were dominated by power in the delta and theta bands, with the hippocampus showing stronger theta activity. During memory trials, activity in both sites transitioned to beta, observed as transient beta clouds in spectrograms and high-magnitude voltages in beta-filtered traces (Fig. 1D).

Raw voltage traces provided clear evidence that brief beta bursts engaged both regions during memory trials (Fig. 1E). Upon trial initiation, the prefrontal cortex exhibited large delta waves accompanied by brief intermittent beta bursts. In the hippocampus, theta oscillations occurred before the trials and continued for ~3 cycles, followed by de novo beta bursts. Paired prefrontal and hippocampal traces showed trial-by-trial alignment, suggesting long-distance coordination.

Coherence analysis was performed to test for beta synchrony between prefrontal-hippocampal sites during memory and maze running (Fig. 1F$_i$). Peak coherence frequencies were 22.34 ± 0.81 Hz during memory and 10.01 ± 0.65 Hz during running. Peak magnitudes were higher during memory compared with running ($t_{(3)} = 13.71$, $p = 9.00 \times 10^{-6}$). Maximum coherence magnitudes ($R^2$) for beta during memory were compared with theta during running as a benchmark interregional coordination strength. Beta coherence magnitudes during memory (memory-related $\beta_{coherence} = 0.37 ± 0.021$) were comparable to theta values during running (running-related $\theta_{coherence} = 0.35 ± 0.01$; $t_{(3)} = 1.30$, $p = 0.24$). A behavior-by-frequency analysis of variance (ANOVA) to compare the overall magnitudes or shapes of coherence distributions in memory vs. running revealed a significant behavior effect ($F_{(1,824)} = 660.42$, $p = 1.27 \times 10^{-99}$) with higher coherence values during memory compared with running; a significant frequency effect ($F_{(102,824)} = 6.36$, $p = 8.05 \times 10^{-50}$), which partially reflected the 1/f noise; and a significant interaction effect ($F_{(102,824)} = 3.38$, $p = 1.31 \times 10^{-20}$). This finding provides support for the notion that prefrontal-hippocampal synchrony shifts between brief beta states during memory and theta states during running. We plotted a difference function (memory-running) to identify the frequencies that synchronize or desynchronize (Fig. 1F$_i$ inset), reinforcing that beta synchrony increased during memory ($t_{(3)} = 21.05$, $p = 2.35 \times 10^{-4}$), while theta trended lower ($t_{(3)} = -2.99$, $p = 0.06$). We analyzed a prior beta and theta bands using area under the curve (AUC) measurements (Fig. 1F$_{ii}$). Beta increased during memory ($t_{(6)} = 4.03$, $p = 0.01$) and theta did not differ significantly between conditions ($t_{(6)} = -0.47$, $p = 0.65$). Lastly, we tested the possibility that beta coherence reflects the stationary state of the animal during memory trials and found that beta coherence was significantly greater during memory trials compared with maze stationary periods on the linear track (Supplementary Fig 4A). These results indicate that the prefrontal-hippocampal synchrony during sequence memory is predominantly driven by transient increases in beta coherence.

To further understand beta activity in prefrontal and hippocampal sites in memory, we plotted the bandpass-filtered traces trial-by-trial (Fig. 1G$_i$, Supplementary Fig 5). Beta in the prefrontal cortex and hippocampus were closely matched in time and amplitude. High-amplitude beta appeared ~100 ms after the trials started. We overlaid prefrontal and hippocampal traces to evaluate cycle-by-cycle changes in amplitudes and phases. With smaller beta amplitudes, synchrony was poor, but with larger amplitudes prefrontal-hippocampal pairs were coupled in both amplitude and phase (Fig. 1G$_{ii}$).

Notably, beta events were bursty rather than continuous; thus, we developed a burst detection algorithm to plot the cumulative density function during trials. Cumulative density functions in both regions were identical (Fig. 1H, $t_{(609)} = 0.020$, $p = 0.98$). Mean burst durations ($t_{(1163)} = 0.51$, $p = 0.61$; Fig. 1I) and latencies to the first burst ($t_{(1018)} = 0.46$, $p = 0.65$; Fig. 1J) were similar.

Burst latencies suggested that prefrontal-hippocampal beta could be involved in the mnemonic aspects of the task. We analyzed prefrontal-hippocampal beta coherence as a function of sequential contexts (InSeq or OutSeq), which requires memory. Perievent

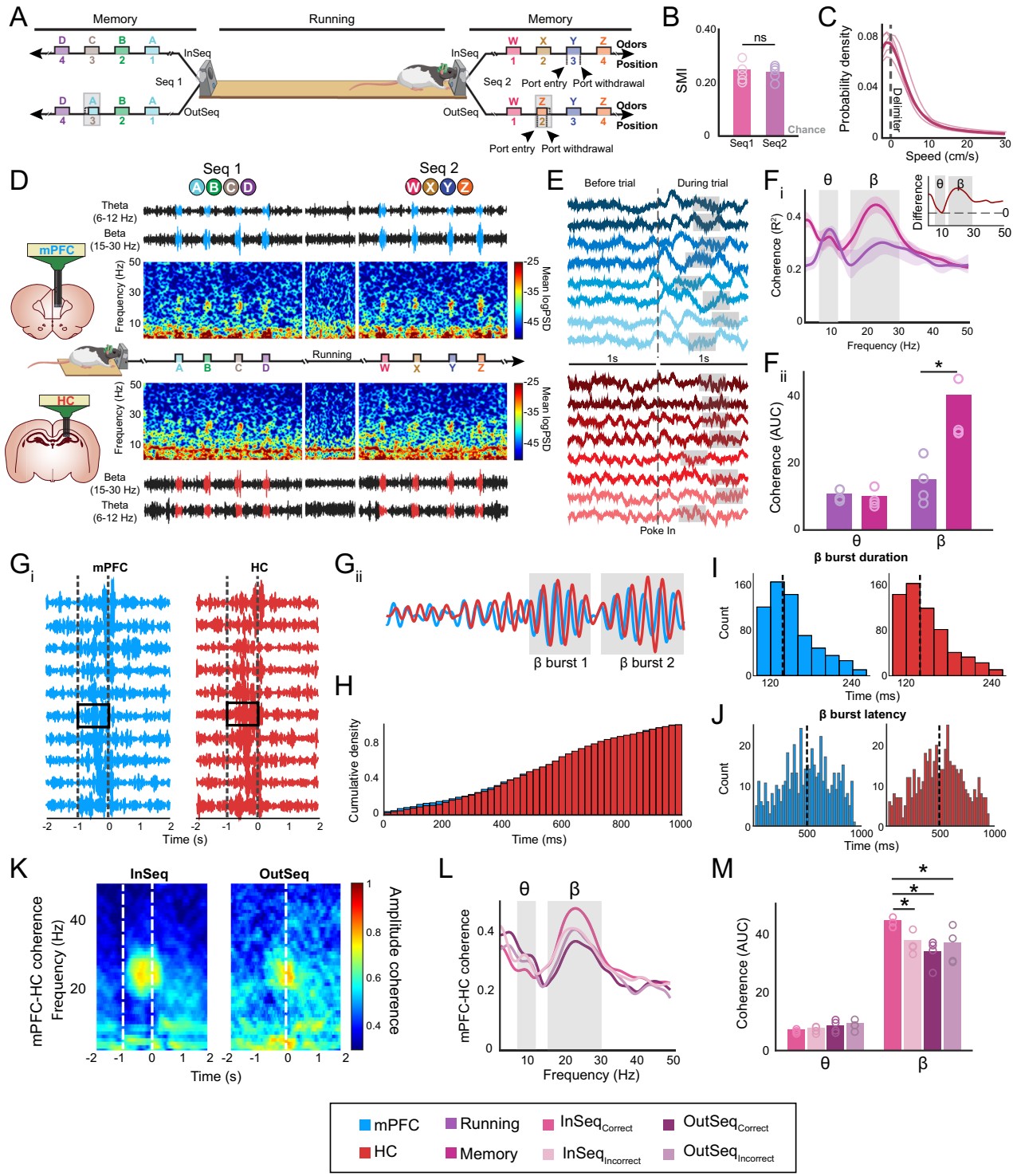

coherograms were aligned to nose-poke withdrawals and averaged across trials ±2 s (Fig. 1K). During InSeq trials, large-magnitude beta clouds were observed in the second half of an odor trial, but during OutSeq trials, the beta activity was weaker and shorter. While this is consistent with prefrontal-hippocampal beta coherence differentiating the sequence contexts, this may be due to differences in the nose-poke times (>1 s InSeq and <1 s OutSeq). Thus, we organized trials by sequential context and accuracy, resulting in 4 conditions (Fig. 1L). While each condition showed high coherence, beta activity in InSeq$_{correct}$ trials had a consistently larger magnitude (similar to CA1 beta[36]). We calculated the AUC (Fig. 1M) and tested differences across conditions. We found that beta activity differed across

trial types ($F_{(3,15)} = 4.24$, $p = 0.03$). Post hoc analysis revealed that the InSeq$_{correct}$ values were higher compared with InSeq$_{incorrect}$ ($p = 0.02$), OutSeq$_{correct}$ ($p = 0.01$), and OutSeq$_{incorrect}$ ($p = 0.02$) trials. If longer nose-poke times alone caused increased beta coherence, we would expect to see more beta activity during OutSeq$_{incorrect}$ trials. To further control for nose-poke time, we compared OutSeq$_{incorrect}$ to InSeq$_{correct}$ trials of the same durations (>1 s) and found a significant difference ($p = 0.016$). Further, OutSeq$_{correct}$ (<1 s) trials were not significantly different from OutSeq$_{incorrect}$ (>1 s) trials ($p = 0.72$), despite having different nose-poke times. Theta activity did not differ significantly across any of the 4 trial types ($F_{(3,15)} = 1.74$, $p = 0.21$). We also averaged perievent power spectral densities sorted by sequential contexts and

**Fig. 1 | Prefrontal-hippocampal system shows transient beta coherence in nonspatial sequence memory.** **A** A linear track was used in which 2 separate 4-odor sequences were presented at opposite ends. Rats had to correctly identify the odor as either InSeq (hold ≥1 s) or OutSeq (hold <1 s). The rat image was modified and reproduced from Jayachandran and Allen[75]. **B** The SMI did not differ significantly between Seq1 and Seq2 ($t_{(4)} = 0.22$, $p = 0.84$; two sample $t$-test). Individual rat performance is indicated by the circles ($n = 5$). **C** Mean running speeds <10 cm/s represents 80% of the data. **D)** Representative spectrogram with corresponding filtered LFP from theta and beta bands in the mPFC (blue) and HC (red) during both sequences interleaved with a short running bout. Brain schematics are originals created using Paxinos and Watson outlines with permission[65]. The rat image was modified and reproduced from Jayachandran and Allen[75]. **E** Sample raw LFP in mPFC and HC. Each rat is indicated with a different shade of color. Beta bursts highlighted in gray. **F$_i$** mPFC-HC coherence differed significantly between running periods (non-memory) and memory (correct odor trials; n = 4). Inset shows the difference between memory and running. **F$_{ii}$** The AUC analysis (two sample $t$-test) shows that the theta coherence was not significantly different between running and memory ($p = 0.65$; $n = 4$), while beta coherence differed significantly between running and memory ($p = 0.01$; $n = 4$). **G$_i$** Bandpass beta-filtered sample

trials from mPFC and HC shows closely matched high amplitude beta ~100 ms after the poke-in. **G$_{ii}$** A zoomed-in trial shows the bursty properties of beta. **H** The probability of beta burst occurrence in the mPFC and HC did not differ significantly ($p = 0.98$; two sample $t$-test). **I** The mean duration of a beta burst was not significantly different between the mPFC and HC ($p = 0.48$). **J** The latency to the first beta burst was not significantly different between the mPFC and HC ($p = 0.95$). **K** mPFC-HC coherence separated based on sequential context (InSeq vs OutSeq). **L** Coherence between InSeq and OutSeq trials separated based on accuracy. **M** InSeq$_{correct}$ Beta AUC was significantly higher compared to InSeq$_{incorrect}$ ($p = 0.02$), OutSeq$_{correct}$ ($p = 0.01$), and OutSeq$_{incorrect}$ ($p = 0.02$). Theta AUC was not significantly different across the 4 trial types ($p = 0.21$; $n = 4$; one-way ANOVA with Bonferroni correction). Abbreviations: mPFC, medial prefrontal cortex (Blue); HC, hippocampus (Red); InSeq, in-sequence; OutSeq, out-of-sequence; SMI, sequence memory index; Seq, sequence; LFP, local field potential; Running (Purple), Memory (Pink); AUC, area under the curve; InSeq$_{correct}$, in-sequence correct (Dark Pink), InSeq$_{incorrect}$, in-sequence incorrect (Light Pink), OutSeq$_{correct}$, out-of-sequence correct (Dark Purple), OutSeq$_{incorrect}$, out-of-sequence incorrect (Light Purple); θ, theta; β, beta; ns, not significant. All data are represented as mean ± SEM; *$p < 0.05$; ns, not significant. Source data are provided as a Source Data file.

---

accuracy in prefrontal and hippocampal sites individually, which demonstrated that the local beta activity at each site was related to memory (Supplementary Fig 6A).

## Beta in RE during memory

The thalamic RE may drive prefrontal-hippocampal beta coherence based on both its anatomical connectivity[11,12,16,17,20] and role in memory[3]. Therefore, we next recorded from RE and hippocampal sites. Rats demonstrated strong and steady memory in both sequences ($t_{(3)} = -0.03$, $p = 0.98$; Fig. 2A, see Supplementary Fig 2B). We first looked at the overall spectrograms of the RE. Figure 2B shows a representative spectrogram from RE during sequence 1, sequence 2, and maze running. Hippocampal recordings in RE-hippocampal rats appeared identical to those of the prefrontal-hippocampal rats. In RE, spectrograms were dominated by sporadic periods of high delta and theta activity between trials, and enormous beta activity during memory trials. RE beta repeated across trials. Beta-filtered voltage traces exhibiting the highest beta amplitudes were time-locked to memory trials (Fig. 2B).

We performed a coherence analysis on RE-hippocampal sites, sorting memory and maze running (Fig. 2C$_i$). Aggregate RE-hippocampal coherence values were calculated and plotted across frequencies. Peak coherence frequencies were 22.09 ± 0.54 Hz during memory and 8.26 ± 0.21 Hz during running. Peak magnitudes were higher during memory compared with running ($t_{(3)} = 23.85$, $p = 3.57 \times 10^{-7}$). Beta coherence magnitudes during memory (memory-related $\beta_{coherence} = 0.42 \pm 0.08$) were comparable to theta magnitudes during running (running-related $\theta_{coherence} = 0.40 \pm 0.06$; $t_{(3)} = 0.14$, $p = 0.89$). Next, we ran a behavior-by-frequency ANOVA, which revealed a behavior effect ($F_{(1,824)} = 448.92$, $p = 2.61 \times 10^{-75}$) with higher coherence values during memory compared with running, a frequency effect ($F_{(102,824)} = 4.28$, $p = 8.23 \times 10^{-30}$), and a behavior-by-frequency interaction ($F_{(102,824)} = 9.28$, $p = 1.61 \times 10^{-74}$). Thus, RE-hippocampus sites showed an increase in overall synchrony, driven by beta activity during memory and theta activity during maze running. We plotted a difference function (memory-running), which showed that beta synchrony increased during memory ($t_{(3)} = 3.86$, $p = 0.03$) while theta was not different ($t_{(3)} = -2.12$, $p = 0.13$; Fig. 2C$_i$ inset). Lastly, we analyzed the AUC for beta and theta activity (Fig. 2C$_{ii}$). The beta AUC increased during memory compared with running ($t_{(6)} = 9.48$, $p = 7.8 \times 10^{-5}$), but the theta AUC did not differ between memory and running ($t_{(6)} = -1.79$, $p = 0.12$). We compared the beta coherence between memory and maze stationary periods, which demonstrated that memory trials had significantly more beta coherence compared with non-memory−related stationary behaviors (Supplementary Fig 4B).

Raw voltage traces in RE-hippocampal sites showed modest delta and theta activity prior to trial initiation, followed by large beta bursts in the RE after trial initiation (and a large delta wave in RE). Unlike the beta activity seen in prefrontal-hippocampal sites, RE beta bursts were stronger and had a clear oscillatory nature. Further, the beta onset latencies in the RE were earlier than those in the prefrontal cortex and hippocampus (Fig. 2D).

If the RE were driving prefrontal-hippocampal beta synchrony, bursts in the RE would occur before those in the hippocampus. To test this, we plotted beta bandpass-filtered traces in each structure on a trial-by-trial basis (Fig. 2E$_i$, Supplementary Fig 5). In each trial, we observed high-amplitude beta bursts in the RE that were much larger and occurred earlier than those in the hippocampus. We overlaid traces to examine cycle-by-cycle changes in the amplitude and phase (Fig. 2E$_{ii}$). Early in the trial, the phase and amplitudes were modestly related, whereas later in the trial (e.g., beta-burst1 and beta-burst2), the amplitudes were tightly coupled.

We then quantified beta burst durations and latencies in the RE and hippocampus. The overlaid cumulative density function of the beta bursts in each region showed that bursts occurred significantly earlier during a trial in the RE than in the hippocampus ($t_{(859)} = 3.62$, $p = 3.17 \times 10^{-4}$; Fig. 2F). Additionally, the average duration of a beta burst was longer in the RE (156.99 ± 1.39 ms) compared with hippocampus (139.75 ± 1.08 ms; $t_{(1582)} = 9.92$, $p = 1.00 \times 10^{-4}$; Fig. 2G), while the beta burst latency in the RE was earlier than that in the hippocampus (RE: 409.35 ± 7.30 ms; hippocampus: 449.80 ± 9.18 ms; $t_{(1419)} = 3.47$, $p = 5.00 \times 10^{-4}$; Fig. 2H).

As with the prefrontal-hippocampal sites, we analyzed RE-hippocampal coherence as a function of sequential context. Perievent coherograms showed strong beta in InSeq compared with OutSeq trials (Fig. 2I). We plotted aggregate coherence sorted by sequential context and accuracy (Fig. 2J). While theta did not differ significantly across the 4 trial types ($F_{(3,15)} = 0.27$, $p = 0.84$), beta did differ significantly (Fig. 2K; $F_{(3,15)} = 4.59$, $p = 0.02$). Post-hoc analysis showed that InSeq$_{correct}$ trials were more beta coherent compared with OutSeq$_{correct}$ trials ($p = 0.01$). Additionally, beta coherence was lower during OutSeq$_{correct}$ trials than during OutSeq$_{incorrect}$ trials ($p = 0.01$). No significant differences were detected between InSeq$_{correct}$ and OutSeq$_{incorrect}$ trials, which had the same hold time ($p = 0.97$). Lastly, InSeq$_{correct}$ trials were not significantly different from InSeq$_{incorrect}$ trials ($p = 0.27$). Beta was consistently higher on the longer nose-poke times; however, and therefore we cannot conclude that coherence is related to mnemonic content in RE-hippocampal pairs. Supplementary Fig 6B shows the perievent spectrograms for each brain region, with only RE showing significant beta power on OutSeq$_{incorrect}$ trials. Thus,

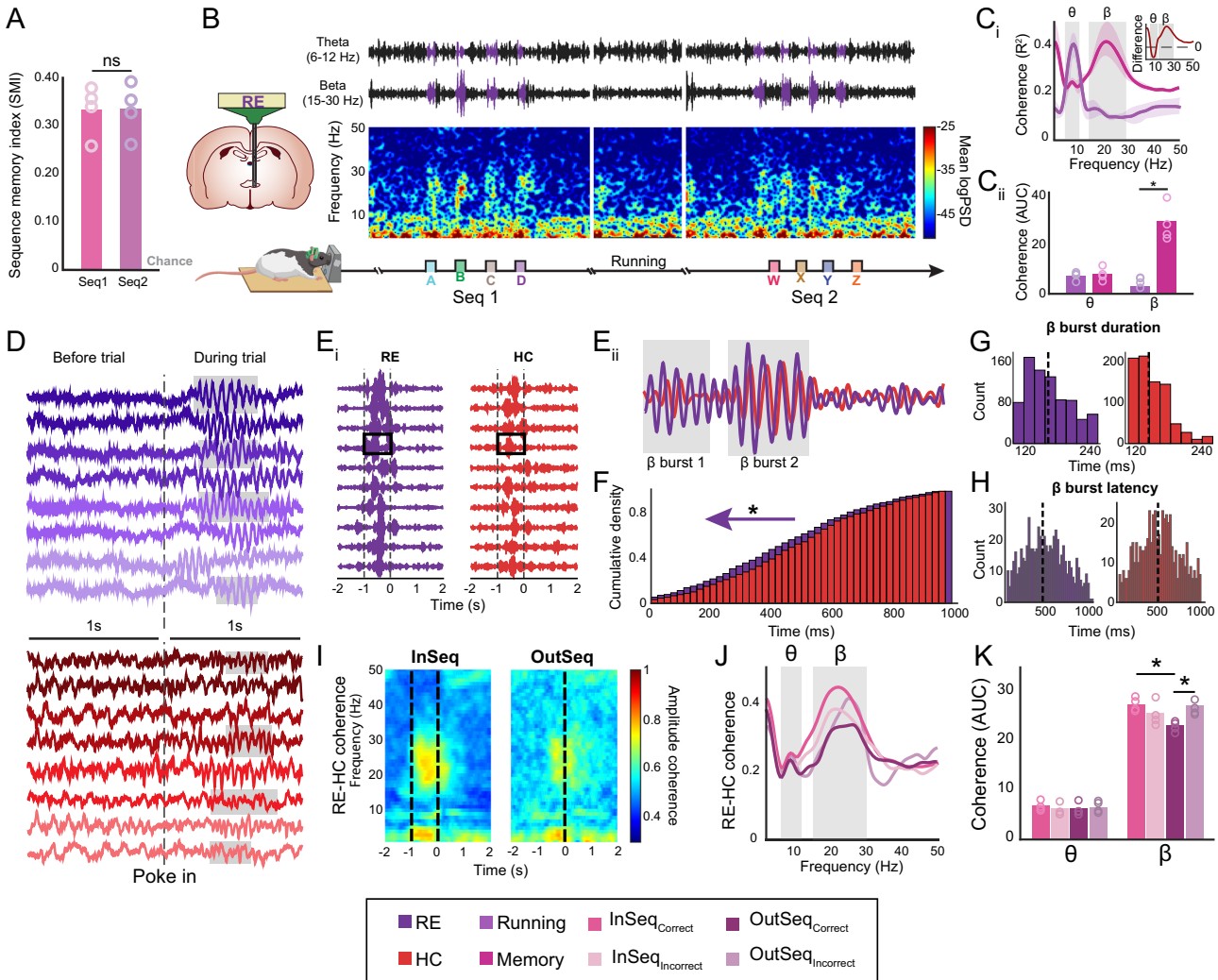

**Fig. 2 | Nonspatial sequence memory induces beta bursts in the RE. A** The SMI did not significantly differ between Seq1 and Seq2 ($t_{(3)} = -0.03$, $p = 0.98$; two sample *t*-test). Circles indicate individual rat performance ($n = 4$). The rat image was modified and reproduced from Jayachandran and Allen[75]. **B** Representative spectrogram with corresponding filtered LFP from theta and beta bands in the RE (purple) during both sequences with a sample running bout in between. High-power beta is aligned to the odor sampling period. Brain schematics are originals created using Paxinos and Watson outlines with permission[65]. **C_i** RE-HC coherence was significantly different between running periods (non-memory) and memory (correct odor trials). Inset shows difference (memory−running; $n = 4$). **C_ii** The AUC analysis (two sample *t*-test) shows that theta coherence was not significantly different between running and memory ($p = 0.12$; $n = 4$), while beta coherence differed significantly between running and memory ($p = 7.8 \times 10^{-5}$; $n = 4$). **D** Sample raw LFP in the RE (purple) and HC (red). Each rat is indicated with a different shade of color. Beta bursts are highlighted in gray. **E_i** Bandpass beta-filtered sample trials from the RE and HC shows that RE beta occurred earlier than HC beta. **E_ii** A zoomed-in trial shows the bursty properties of beta with RE beta occurring earlier in the trial. **F** The probability of RE and HC beta

burst occurrence differed significantly, with that in the RE occurring earlier (see purple arrow; $p = 3.17 \times 10^{-4}$; two sample *t*-test). **G** The mean duration of a beta burst was significantly longer in the RE than in the HC ($p = 5.98 \times 10^{-24}$). **H** The latency to the first beta burst was significantly earlier in the RE than in the HC ($p = 0.01$). **I** RE-HC coherence separated based on sequential context (InSeq vs OutSeq). **J** Coherence between InSeq and OutSeq trials were separated based on accuracy. **K** The InSeq_correct beta AUC was significantly higher compared to the OutSeq_correct ($p = 0.01$), and the OutSeq_correct beta AUC was significantly lower than OutSeq_incorrect ($p = 0.01$) beta AUC. Theta AUC was not significantly different across the 4 trial types ($p = 0.84$; $n = 4$; one-way ANOVA with Bonferroni correction). Abbreviations: RE, nucleus reuniens (Purple); HC, hippocampus (Red), SMI, sequence memory index; Seq, sequence; LFP, local field potential; Running (Purple); Memory (Pink) AUC, area under the curve; InSeq, in-sequence; OutSeq, out-of-sequence; InSeq_correct, in-sequence correct (Dark Pink), InSeq_incorrect, in-sequence incorrect (Light Pink), OutSeq_correct, out-of-sequence correct (Dark Purple), OutSeq_incorrect, out-of-sequence incorrect (Light Purple); θ, theta; β, beta; ns, not significant. All data are represented as mean ± SEM; *$p < 0.05$; ns, not significant. Source data are provided as a Source Data file.

the RE may engage prefrontal-hippocampal memory networks, but by itself is driven by the behavioral dynamics inherent in the task (such as the poking behavior).

## Timing of beta across the memory network
To directly compare relative amplitudes and time courses for beta and theta across all 3 brain regions, we took the upper envelope from the

beta- and theta-filtered voltage traces, and z-normalized the output (Fig. 3A). These envelopes were averaged across trials (InSeq_correct only to match nose-poke times) and reflect a combination of the probability densities of bursts and changes in amplitudes. Beta starts rising just prior to the onset of a trial and earlier in RE than in the prefrontal cortex and hippocampus. Prefrontal cortex and hippocampus beta activity simultaneously increased only after the trial started (Fig. 3B).

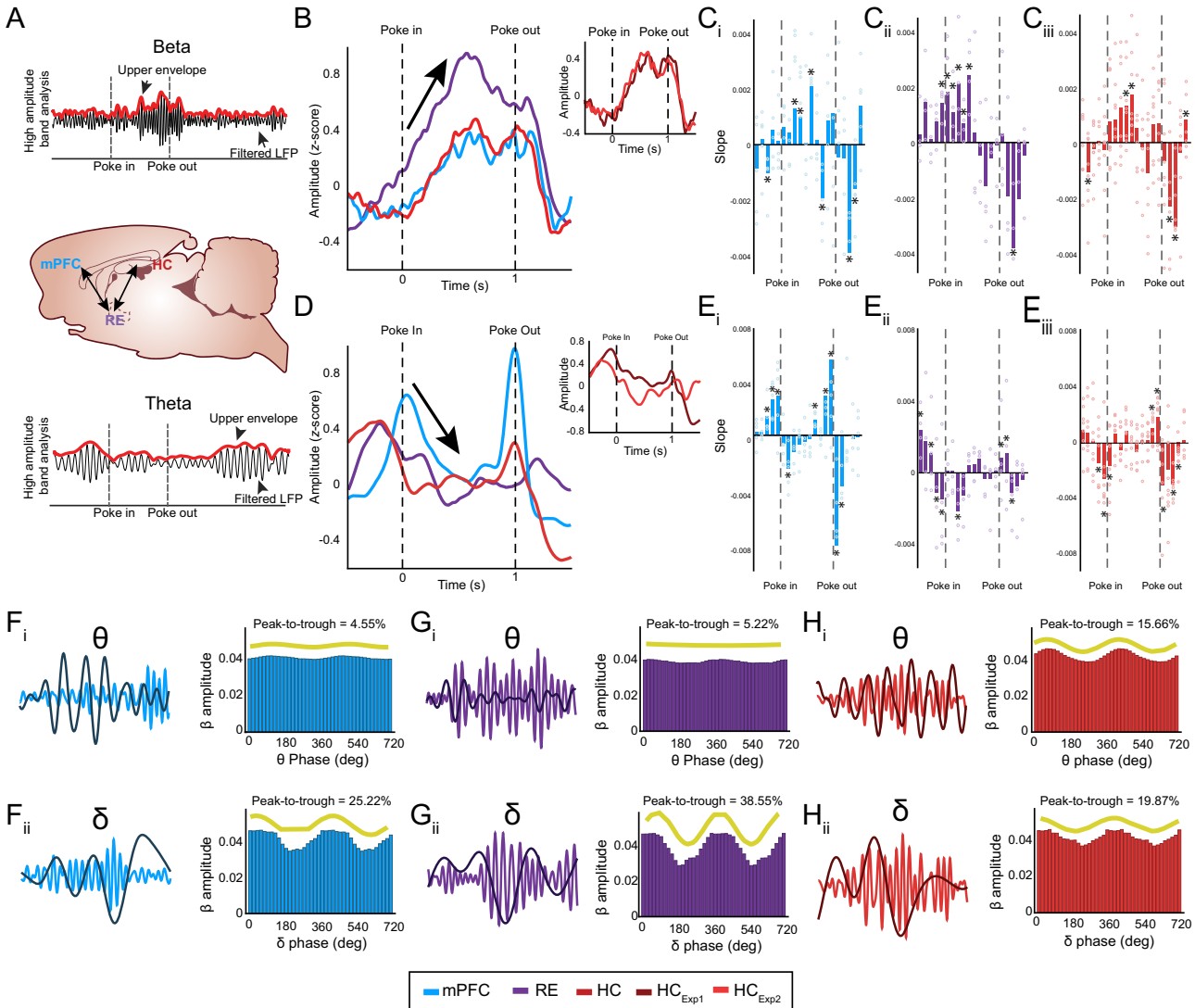

**Fig. 3 | Beta amplitudes rise earlier and stronger in RE and then concurrently in prefrontal cortex and hippocampus.** **A** Sample of upper envelope analysis for beta and theta. Brain schematics are originals created using Paxinos and Watson outlines with permission[65]. **B** Beta amplitudes in the RE start rising just prior to trial onset (arrow). mPFC and HC beta simultaneously rises after the trial. Inset plot shows no significant difference between HC recordings. **$C_i$** Calculated slopes show that mPFC beta significantly rose after poke-in and then decreased after poke-out ($p = 0.05$; $p = 0.01$; $p = 4.0 \times 10^{-3}$; $p = 0.04$; $p = 0.021$; $p = 0.01$; $p = 0.05$; $n = 5$; one sample $t$-test). **$C_{ii}$** RE beta slopes significantly rose for poke-in and continued to rise during the trial ($p = 0.01$; $p = 0.01$; $p = 0.01$; $p = 0.04$; $p = 0.05$; $p = 0.02$; $p = 0.05$; $n = 4$; one sample $t$-test). **$C_{iii}$** HC beta slopes show similar patterns to mPFC ($p = 0.05$; $p = 0.03$; $p = 0.01$; $p = 0.03$; $p = 0.01$; $p = 0.05$; $n = 8$; one sample $t$-test). **D** Theta amplitudes in RE and HC increase before a trial followed by an increase in the mPFC at the start of the trial. During the trial, there is a decrease in theta (arrow). Inset plot shows no significant difference between HC recordings. **$E_i$** Calculated slopes show that mPFC theta significantly rose before the start and end of a trial, while it decreased during the trial ($p = 0.02$; p = 0.01; $p = 0.02$; $p = 0.02$; $p = 0.01$; $p = 2.0 \times 10^{-3}$; $p = 0.01$; $p = 0.02$; $p = 0.04$; $n = 5$; one sample $t$-test). **$E_{ii}$** RE theta slopes significantly rose before poke-in and decreased during the trial

($p = 0.03$; $p = 0.05$; $p = 0.02$; $p = 0.01$; $p = 0.04$; $p = 0.05$; $p = 1.0 \times 10^{-3}$; $p = 0.03$; $n = 4$; one sample $t$-test). **$E_{iii}$** HC theta slopes exhibited patterns similar to those of the RE and mPFC ($p = 0.01$; $p = 0.02$; $p = 0.1$; $p = 0.01$; $p = 0.01$; $p = 0.03$; $p = 1.06 \times 10^{-4}$; $p = 0.04$; $n = 8$; one sample $t$-test). **$F_i$, $F_{ii}$** Sample of mPFC beta voltage (blue) superimposed with theta or delta voltage (dark blue). **$F_i$** Phase-amplitude plot from a rat showing beta amplitude was weakly modulated by theta phase. **$F_{ii}$** Phase-amplitude plot showing beta amplitude was more modulated by delta phase. **$G_i$, $G_{ii}$** Sample of RE beta voltage (purple) superimposed with theta and delta voltages (dark purple). **$G_i$** Phase-amplitude plot showing that beta amplitude was weakly modulated by theta phase. **$G_{ii}$** Phase-amplitude plot from a rat showing that beta amplitude was more modulated by delta phase. **$H_i$, $H_{ii}$** Sample of HC beta voltage (red) superimposed with theta voltage (dark red). **$H_i$** Phase-amplitude plot showing that beta amplitude was modulated by theta phase. **$H_{ii}$** Phase-amplitude plot from a rat showing that beta amplitude was modulated by delta phase. Yellow waveforms represent the fitted curves to the amplitude–phase distributions. Abbreviations: mPFC, medial prefrontal cortex (Blue); RE, nucleus reuniens (Purple); HC hippocampus (Red), Exp experiment, $HC_{Exp1}$ (Dark Red) $HC_{Exp2}$ (Light Red), θ theta, δ delta, β beta. All data are represented as mean ± SEM; *$p < 0.05$; ns, not significant. Source data are provided as a Source Data file.

We compared the hippocampal recordings from the 2 experiments and found no significant difference between regions, but a significant difference across time (region: $F_{(1,3)} = 0.28$, $p = 0.63$; time: $F_{(19,57)} = 13.91$, $p = 5.61 \times 10^{-15}$; region-x-time: $F_{(19,57)} = 0.36$, $p = 0.99$; Fig. 3B inset). We combined the recordings from the 2 experiments and ran a region-x-time ANOVA, which revealed a main effect of region ($F_{(2,16)} = 33.24$, $p = 1.00 \times 10^{-3}$), a main effect of time ($F_{(19,57)} = 28.91$,

$p = 1.71 \times 10^{-22}$), and a region-x-time interaction ($F_{(38,114)} = 2.44$, $p = 1.48 \times 10^{-4}$). This pattern indicates that (1) poking behaviors initiate beta in RE, and (2) RE drives prefrontal-hippocampal beta coherence. To provide further evidence, we calculated the slope in 100 ms bins and tested for an increase in beta activity before, during, and after a trial (Fig. 3$C_{i-iii}$). This analysis showed that RE beta increased before the start of a trial, whereas prefrontal cortex and hippocampus beta was

low before a trial (1 sample *t*-test, *p* < 0.05). Interestingly all 3 brain regions peaked at the same time, ~500 ms after the trial onset, and decreased slightly or remained steady (near zero) for the remainder of the trial. Beta returned to baseline levels in all brain regions ~100 ms after the trial offset.

The time course of theta was quite different from that of beta. Theta showed brief increases in amplitude in the RE and hippocampus before a trial, followed by an increase in amplitude in the prefrontal cortex around the start of the trial (Fig. 3D). Theta decreased in all regions in the first half of a trial and remained near zero until the end of the trial, marked by an increase in theta in the prefrontal cortex and hippocampus followed by the RE. We compared hippocampal recordings from the 2 experiments and found no significant difference between regions, but a significant difference across time (region: $F_{(1,3)} = 3.19$, *p* = 0.17; time: $F_{(19,57)} = 7.52$, *p* = $1.37 \times 10^{-9}$; region-x-time: $F_{(19,57)} = 1.69$, *p* = 0.06; Fig. 3D inset). We combined data from the 2 experiments and ran a region-x-time ANOVA, which revealed a main effect of region ($F_{(2,6)} = 0.08$, *p* = 0.92), a main effect of time ($F_{(19,57)} = 3.81$, *p* = $4.60 \times 10^{-5}$), and a region-x-time interaction ($F_{(38,114)} = 2.51$, *p* = $9.30 \times 10^{-5}$). We next calculated the slope in 100 ms bins and tested whether theta activity increased before, during, and after a trial (Fig. 3E$_{i-iii}$). This analysis showed that theta increased in the prefrontal cortex, RE, and hippocampus before the start of a trial and around poke-out (1 sample *t*-test, *p* < 0.05). All 3 brain regions peaked and quickly decreased (poke-in) and remained steady (near zero) during the trial. Theta peaked in all brain regions close to the end of the trial. These observations suggest that theta influences the network during the onset and offset, rather than during memory trials.

To directly test for beta-theta relationships, we ran a phase-amplitude coupling analysis during trials (Fig. 3F–H). We estimated instantaneous phases during memory trials in theta and delta and plotted the beta envelope (amplitude) as a function of the theta or delta phase. We then fitted the plots to sine waves and calculated significant fits (*p* < 0.05).

We calculated the percentage peak-to-trough distance by subtracting the minimum beta amplitude (trough) from the maximum beta amplitude (peak). We ran a region-x-frequency ANOVA here which revealed a main effect of frequency ($F_{(1,34)} = 32.50$, *p* = $4.00 \times 10^{-6}$). The theta relationships were lower (prefrontal: 4.95 ± 0.01; RE: 5.19 ± 0.01; hippocampus: 11.22 ± 0.02; Fig. 3F$_i$, G$_i$, H$_i$) than the delta relationships (prefrontal: 19.45 ± 0.02; RE: 17.84 ± 0.06; hippocampus: 23.86 ± 0.03; Fig. 3F$_{ii}$, G$_{ii}$, H$_{ii}$). Beta was coupled to delta in all 3 regions. Beta amplitudes were coupled to the theta amplitude mostly in the hippocampus (Supplementary Fig 7). This result is consistent with the observed memory-related prefrontal-hippocampal and RE-hippocampal coherence curves (Supplementary Fig 4).

Altogether, the results suggest complex dynamics between beta and theta occur during sequence memory. Theta increased at the onset and offsets of trials, whereas beta was highest during the trial–increasing in RE first and then simultaneously in the prefrontal cortex and hippocampus. All regions exhibited varying degrees of beta-amplitude coupled to the delta phase, and only the hippocampus showed beta-amplitude coupled to the theta phase.

## RE drives network beta activity

The latency analysis results were consistent with a model that the RE drives prefrontal-hippocampal synchrony, thus we next tested the causal capability of the RE to generate prefrontal-hippocampal coherence. We used an optogenetic viral approach to stimulate retrogradely-labeled neurons in the RE that project to the hippocampus. While stimulating, we recorded LFP activity in the prefrontal cortex and hippocampus (Fig. 4A$_i$, Supplementary Fig 1C) as rats freely explored an open-field and had no memory demands (*n* = 10). We confirmed expression patterns histologically in RE (ChR2-rats and nonChR2-rats; Fig. 4A$_{ii}$) as well as electrode and optrode placements

(Supplementary Fig 1C). Blue light was delivered via an optic fiber positioned above the RE with an LED source in pulse or sine wave patterns across different frequencies (Fig. 4B). The use of multiple frequencies helped determine if a specific stimulus pattern was required to induce prefrontal-hippocampal synchrony.

We first assessed whether blue light affected LFP activity in prefrontal-hippocampal recording sites across days and groups (Fig. 4C, Supplementary Fig 8). In ChR2-rats, blue light delivered to the RE evoked strong monosynaptic responses in the hippocampus, consistent with other reports[44]. We never observed evoked potentials in nonChR2-rats. This demonstrated the specificity of the effect of ChR2 activation in the RE.

We measured the latency of negative and positive peak occurrences in hippocampal voltage traces following the onset of each light pulse. During the 60 ms pulse, the frequency of the negative peaks was highest in the 10–15 ms bin (first peak) and then at 30–35 ms (second peak). The positive peaks appeared between 15–20 ms (first peak) and 50–55 ms (second peak; Fig. 4C inset). The modal latency of negative peaks during pulses was 12 ms, indicating that optogenetic manipulations of the RE elicited monosynaptic responses in the hippocampus, consistent with the anatomy and physiology of RE projections[12,16,20].

In ChR2-rats, sine wave stimulation caused changes to the voltage activity recorded in the prefrontal cortex and hippocampus (Supplementary Fig 8C). Overall, sine wave blue-light stimulation triggered rapid down-state transitions at the rise of sine waves and rapid up-state transitions with the fall of sine waves, generating a weak sinusoidal rhythm in prefrontal and hippocampal traces.

We assessed whether the RE induced prefrontal-hippocampal coherence at larger time scales (seconds-to-minutes). Beta was only commonly seen in traces from ChR2-rats compared with nonChR2-rats after light stimulation (Fig. 4D). This suggested RE activation was sufficient to increase beta in the prefrontal cortex and hippocampus.

Next, we tested whether RE neurons could drive beta and theta coherence in the prefrontal-hippocampal network. We assessed coherence 1 min before, during, and 1 min after stimulation. Prefrontal-hippocampal coherence curves were largely independent of the stimulation pattern (Supplementary Fig 9A–C). Therefore, we collapsed all frequencies of pulse and sine stimulations into mean group coherences (Fig. 4E$_{i-iii}$). Prefrontal-hippocampal coherence$_{before}$ stimulation was similar in ChR2-rats and nonChR2-rats (Fig. 4E$_i$) whereas prefrontal-hippocampal coherence$_{during}$ drastically changed in ChR2-rats compared with nonChR2-rats. Specifically, theta coherence was reduced and beta coherence was increased. The beta coherence increase was a large effect (~ 1.8 z-scores at the peak; Fig. 4E$_{ii}$). Interestingly, beta coherence in ChR2-rats remained high for at least 1 min after stimulation for both pulse and sine wave stimulations. This effect was not observed in nonChR2-rats (Fig. 4E$_{iii}$).

Notably, a weak residual beta coherence was also observed in pulse and sine wave stimulation during the 1 min-before window, likely stemming from the fact that stimulated blocks (collapsed frequencies) were preceded by other stimulated blocks during the session (Fig. 4E$_i$). Coherence$_{before}$ was significantly different from coherence$_{during}$ and coherence$_{after}$ in ChR2-rats (*p* < 0.05), demonstrating that RE drove beta in the network. Additionally, a 20 Hz sine wave caused a large and narrow 20 Hz prefrontal-hippocampal coherence peak (Fig. 4E$_{ii}$ inset) that did not occur for other frequencies (Supplementary Fig 9), suggesting that the network was resonant within the beta range.

Next, we calculated the AUC for delta, theta and beta across the three different time-windows and between groups (Fig. 4F, G). Time had a significant effect on delta coherence (Fig. 4E; $F_{(2,36)} = 3.36$, *p* = 0.046) with significant reductions between delta$_{before}$ and delta$_{during}$ (*p* = 0.048), but no difference between delta$_{before}$ and delta$_{after}$ (*p* = 0.256), and between delta$_{during}$ and delta$_{after}$ (*p* = 1.00). We also found a main effect of group ($F_{(2,36)} = 20.08$, *p* = $1.00 \times 10^{-6}$) with significant differences between delta$_{control}$ and delta$_{pulse}$

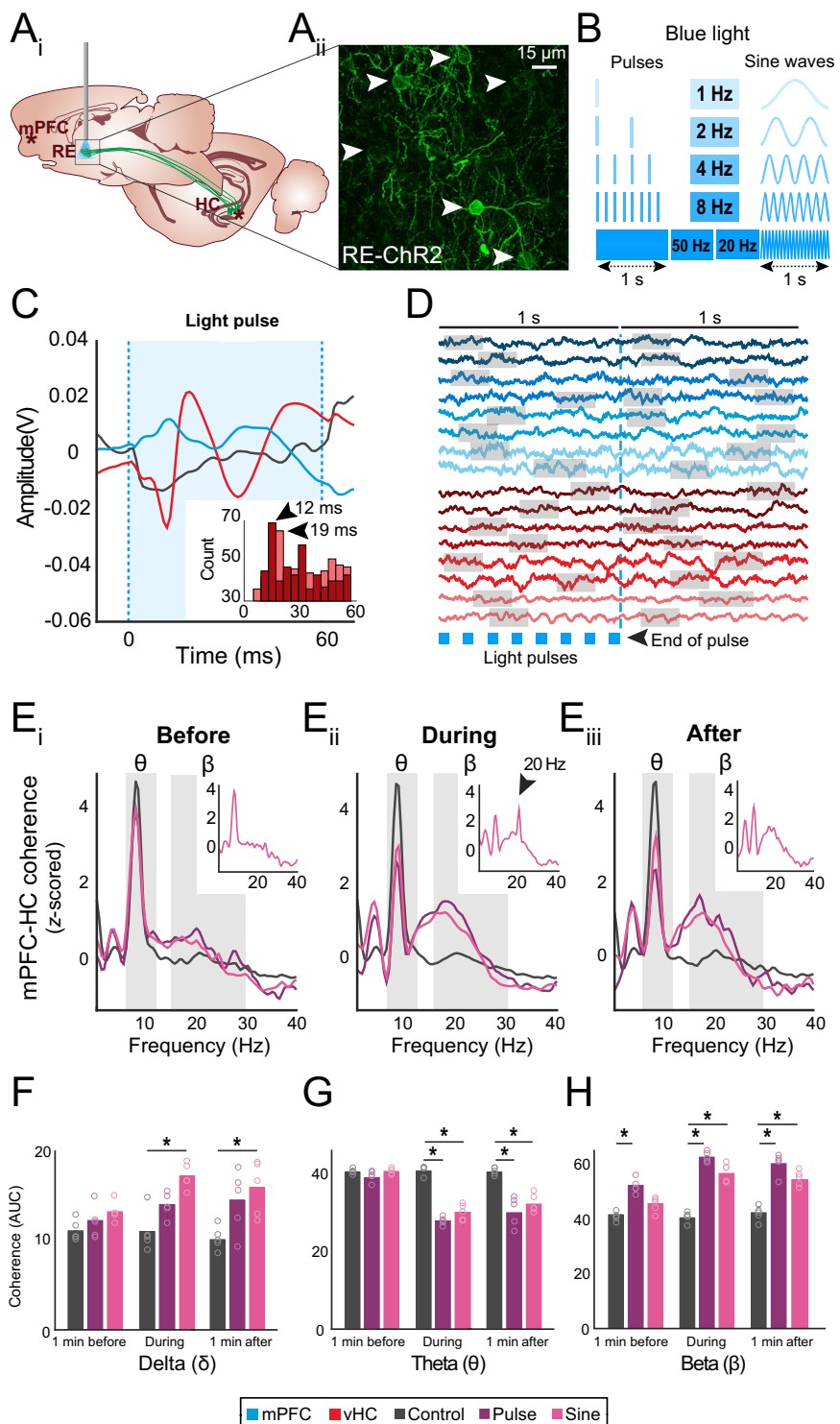

($p = 0.002$), in delta$_{control}$ and delta$_{sine}$ ($p = 1.00 \times 10^{-6}$), and between delta$_{pulse}$ and delta$_{sine}$ ($p = 0.049$). There was no significant time-x-group interaction ($F_{(4,36)} = 1.80$, $p = 0.150$). These results indicate that RE activation increases prefrontal-hippocampal coherence in delta for ChR2-rats during stimulations. Time had a significant effect on theta coherence (Fig. 4F; $F_{(2,36)} = 58.08$, $p = 5.39 \times 10^{-12}$) with significant reductions between theta$_{before}$ and theta$_{during}$ ($p = 1.30 \times 10^{-11}$), theta$_{before}$ and theta$_{after}$ ($p = 2.50 \times 10^{-9}$), but no difference between theta$_{during}$ and theta$_{after}$ ($p = 0.19$). We also found a main effect of group ($F_{(2,36)} = 75.07$, $p = 1.43 \times 10^{-13}$) with significant reductions in theta$_{control}$ and theta$_{pulse}$ ($p = 2.14 \times 10^{-13}$), in theta$_{control}$ and theta$_{sine}$

($p = 3.87 \times 10^{-10}$), but no difference between theta$_{pulse}$ and theta$_{sine}$ ($p = 0.72$). There was also a significant time-x-group interaction ($F_{(4,36)} = 15.18$, $p = 2.32 \times 10^{-7}$), indicating that RE activation overall decreases prefrontal-hippocampal coherence in theta for ChR2-rats during and after stimulation. Time also had a significant effect on beta coherence (Fig. 4G; $F_{(2,36)} = 27.24$, $p = 6.25 \times 10^{-8}$), with significant increases in beta coherence between beta$_{before}$ and beta$_{during}$ ($p = 1.77 \times 10^{-7}$), beta$_{before}$ and beta$_{after}$ ($p = 3.00 \times 10^{-6}$), but no difference between beta$_{during}$ and beta$_{after}$ ($p = 1.00$). We also found a main effect of group ($F_{(2,36)} = 149.82$, $p = 3.53 \times 10^{-18}$) with significant increases in beta$_{control}$ and beta$_{pulse}$ ($p = 2.19 \times 10^{-18}$), in beta$_{control}$

**Fig. 4 | Optogenetic stimulation of RE drives beta coherence in prefrontal-hippocampal system. $A_i$** Bilateral injections of retrograde pAAV-Syn-ChR2(H134R)-GFP (experimental) or retrograde AAV-CAG-GFP (controls) were delivered in the vCA1 for the retrograde expression of RE neurons. Optic fiber implanted above the RE. Brain schematics are originals created using Paxinos and Watson outlines with permission[65]. **$A_{ii}$** Confocal image showing retrograde expression of ChR2 in RE neurons (white arrows). ChR2-expression was confirmed in all experimental rats ($n = 5$). **B** LED-based blue light pulse or sinewave stimulations were used to activate RE-ChR2 transduced rats. **C** Mean evoked responses in vCA1 resulting from 8-Hz stimulations in the RE in a control and an experimental subject. Stimulation of RE-ChR2 expressing neurons resulted in 2 consecutive negative deflections in HC (red). Mild to moderate LFP changes in mPFC activity were also observed (blue) in RE-ChR2 rats compared to controls (black). Pulse and sine stimulations were repeated across 3 different sessions in all rats. **D** Sample mean raw traces in mPFC (blue) and HC (red; 2 samples/experimental rat) showing a predominant beta rhythm 1 s prior and 1 s after the last optogenetic pulse. Gray boxes show high beta activity. **E** z-scored mPFC-HC coherence in RE-ChR2 and controls across 3 epochs: **$E_i$** 1-min before, all rats have similar theta and beta

amplitude values, **$E_{ii}$** during stimulation, marked increases in the beta band and a decrease in theta were observed in RE-ChR2 rats but not controls, **$E_{iii}$** 1-min after stimulation, coherence is similar to **$E_{ii}$**. Notably, 20 Hz sine wave stimulations of RE-ChR2 neurons resulted in a large increase in beta and an additional sharp peak around 20 Hz that surpassed the amplitude of the delta and theta bands (**$E_{ii}$** inset, black arrow); this effect was not seen before or after (**$E_i$ & $E_{iii}$** inset). **F** Delta AUC coherence was significantly different across conditions (control/pulse/sinewave; $p = 1.00 \times 10^{-6}$) and time windows (before, during, after; $p = 0.046$; $n = 5$ per condition; one-way ANOVA with Bonferroni correction). **G** Theta AUC coherence was significantly different across conditions ($p = 1.43 \times 10^{-13}$) and time windows ($p = 5.39 \times 10^{-12}$; $n = 5$ per condition; one-way ANOVA with Bonferroni correction). **H** Beta AUC coherence was significantly different across conditions ($p = 3.53 \times 10^{-18}$) and time windows ($p = 6.25 \times 10^{-8}$; $n = 5$ per condition; one-way ANOVA with Bonferroni correction). All statistical tests are two tailed. AUC area under the curve, ChR2 channelrhodopsin, HC hippocampus (Red), min minute, ms milliseconds, mPFC medial prefrontal cortex (Blue), RE reuniens, vCA1 ventral CA1. Control (Black); Pulse (Purple); Sine (Pink). θ, theta; δ, delta; β, beta. All data are represented as mean ± SEM; $^*p < 0.05$. Source data are provided as a Source Data file.

and beta$_{sine}$ ($p = 1.86 \times 10^{-12}$), and between beta$_{pulse}$ and beta$_{sine}$ ($p = 1.00 \times 10^{-6}$). There was a significant time-x-group interaction ($F_{(4,36)} = 8.15$, $p = 8.7 \times 10^{-5}$), indicating that RE activation overall increased beta coherence in ChR2-rats during and after stimulation. This showed that optogenetic activation of RE significantly increased prefrontal-hippocampal beta coherence, while it decreased theta coherence during and after stimulation. Altogether, these results demonstrate that RE activation is sufficient to drive prefrontal-hippocampal beta coherence while reducing theta coherence, which replicated the prefrontal-hippocampal coherence pattern observed during the sequence memory task.

## Discussion

Our findings revealed that 15–30 Hz beta activity accounted for a majority of the coherence in prefrontal-RE-hippocampal interactions during sequence memory retrieval, and that RE neurons can drive 15–30 Hz beta coherence between the prefrontal cortex and hippocampus in freely-moving animals. We found that prefrontal-hippocampal beta coherence reflected memory content, and not simply the sensory aspects of odor processing or motor behaviors, while RE-hippocampal beta coherence reflected poking behaviors irrespective of the memory condition. This observation suggests a model of episodic retrievals whereby the RE is activated by behavior, and in turn engages prefrontal-hippocampal memory processing modes. Interestingly, beta coherence is observed between the olfactory bulb and hippocampus when there is a flexible use of context-appropriate odor memory[37,40,41], and when the prefrontal cortex is key to resolving interference[56]. One of the more intriguing possibilities here is that beta activity enables episodic buffers in working memory, similar to observations of beta activity in non-human primates[42]. From this perspective, RE-related beta modes may serve as the neurobiological mechanism of an episodic buffer, temporarily maintaining information content stability in the time leading up to a decision[57]. Whether and how beta is related to mechanisms of maintaining spatial processing in addition to sequences remains to be investigated[40]. Future experiments should explore how disrupting or enhancing beta/delta oscillations in real-time by optogenetically stimulating RE-vHC neurons before, during and after memory-specific behaviors (odor stimulus) in a controlled manner (closed-loop control) can impact hippocampal-mPFC coherence. Speculatively, stimulating reuniens inputs during the odor stimulus may boost directed retrieval[2] and delta-beta rhythms, but possibly impair performance if stimulation occurs before or after the odor presentation, which may be more critical for other theta-dominated memory functions[21,58].

Our finding that beta amplitudes strongly couple to delta phases suggests that delta-beta modes support local and long-range prefrontal-hippocampal interactions. Consistent with this idea, beta

bursts were transient and could be described as beat frequencies lasting ~100–240 ms, a pattern similar to amplitude modulations in the delta range[34]. While the cellular-level mechanisms of beta coherence are not known, cross-regional delta-beta couplings initiated by the RE might change the local excitatory-inhibitory tone in the prefrontal cortex and hippocampus through feed-forward inhibition[5]. This would allow the timing of the beta bursts to be shaped by phases of delta similarly to cross-regional theta-gamma couplings[59,60].

Finally, we demonstrated the causal role of the RE in synchronizing prefrontal-hippocampal coherence in the beta range. Previous reports demonstrated that cortico-thalamo-cortical circuits are essentially involved in coordinating activity across distal cortical brain regions[61,62]. Understanding how the delta-beta modes observed here relate to the role of the RE in coordinating prefrontal-hippocampal networks during slow-wave sleep[25] may help to further illuminate the mechanisms how the prefrontal-hippocampal network supports episodic memory.

In conclusion, the present work provides compelling evidence that the RE contributes to memory by driving transient beta synchrony in the prefrontal-hippocampal system, thereby facilitating interactions that underlie memory-based decision making.

## Methods

### Subjects

Nineteen Long-Evans rats (9 males, 10 females) from Charles River Laboratories weighing 250–350 g upon arrival were used. Rats were individually housed and maintained on a 12 h inverse light/dark cycle (lights off at 10 AM). Rats trained on the sequence memory task had *ad libitum* access to food and enrichment, but access to water was limited to 3–5 min each day, depending on how much water they received as a reward during behavioral training (6–9 mL). Rats used in the open field task had ad libitum access to food, enrichment, and water. All training and testing sessions were conducted during the dark phase (active period) of the light cycle. All experimental procedures using animals were conducted in accordance with the Florida International University Institutional Animal Care and Use Committee (FIU IACUC). Although both male and female rats were used, sex was not evaluated as a variable as performance in the sequence memory task does not differ between males and females[63].

### Sequence memory experiments

**Sequence memory task.** The sequence memory task used[3,36,63] involves repeated presentations of odor sequences and requires the rat to determine whether each item (odor) was presented in sequence (InSeq; by holding the nose-poke response for 1 s) or out of sequence (OutSeq; by withdrawing its nose from the port before 1 s). Rats were trained on two sequences, each comprising four distinct odors

(e.g., Seq1: ABCD, Seq2: WXYZ). Each sequence was presented at either end of a linear track maze. Odor presentations were initiated by a nose-poke, and each trial was terminated after the rat either held the nose poke response for >1 s (InSeq) or withdrew its nose-poke response for <1 s (OutSeq). There was a 1 s interval between trials. Water rewards (20 μl; diluted at 1 g of aspartame for every 500 mL of water) were delivered below the odor port after each correct response. Following an incorrect response, a buzzer sound was emitted and the sequence was terminated. Each sequence was presented alternately 50–100 times per session; approximately half the presentations included all items InSeq (ABCD) and half included one item OutSeq (e.g., ABAD, odor A repeated in the 3rd position). Note OutSeq items could be presented in any sequence position except the first position (i.e., sequences always began with an InSeq item). Sequence memory was probed with OutSeq trials (e.g., ABAD; one OutSeq trial randomly presented per sequence).

### Sequence task apparatus

Rats trained on the sequence memory task were tested in a noise-attenuated experimental room. The behavioral apparatus was comprised of a linear track (length, 183 cm; width, 10 cm; height, 43 cm) with walls angled outward at 15° and nose ports at each end through which repeated deliveries of multiple distinct odors could be presented. Photobeam sensors were used to detect nose port entries. Each nose port was connected to an odor delivery system (Med Associates). Odor deliveries were initiated by a nose-poke entry and terminated either when the rat withdrew before 1 s, or after 1 s had elapsed. Water ports were positioned under each nose port for reward delivery. Timing boards (Plexon) and digital input/output devices (National Instruments) were used to measure all event times and control the hardware. All aspects of the task were automated using custom scripts using core MATLAB functions (MathWorks R2016a). A 256-channel Omniplex D with video tracking and Cineplex behavior software (Plexon) were used to interface with the hardware in real time and record behavioral data. Odors were organic odorants contained in glass jars (A: 1-octanol; B: (-) - limonene; C: I-menthone; D: isobutyl alcohol; W: acetophenone; X: (1 S) - (-) – beta pinene; Y: L (-) - carvone; X: 5-methyl-2- 890 hexanone) that were volatilized with nitrogen air (flow rate, 2 L/min) and diluted with ultrapure air (flow rate, 1 L/min). To prevent cross-contamination, separate Teflon tubing lines were used for each odor. These lines converged into a single channel at the bottom of the odor port. In addition, a vacuum located at the top of the odor port provided constant negative pressure to quickly evacuate odor traces with a matched flow rate.

### Sequence memory task training

Naive rats were initially trained in a series of incremental stages over 20–30 weeks. Each rat was trained to poke and hold its nose in an odor port to receive a water reward. The minimum required nose-poke duration started at 50 ms and was gradually increased (in 15 ms increments) until the rat reliably held the nose-poke position for 1.2 s for ~70% of the time over three sessions (75–100 nose-pokes per session). The rats were then habituated to odor presentations in the port (odor A and W, then odor sequences AB and WX) and each rat was required to maintain its nose-poke response for 1 s to receive a reward. The rats were then trained to identify InSeq and OutSeq items. Rats were initially trained on a two-item sequence in equal proportions. The correct response to the first odor was to hold the nose-poke for 1 s (Odor A and Odor W was always the first item). For the second odor, rats were required to determine whether the item was InSeq (hold for 1 s to receive reward) or OutSeq (withdraw before 1 s to receive a reward). After reaching criterion on the two-item sequence, the number of items per sequence was increased to three and four in successive stages (criterion: ~60% correct across all individual odor presentations over three sessions). After reaching criterion

performance on the two four-odor sequences (60% correct on both InSeq and OutSeq items), rats underwent surgery to implant chronic recording electrodes.

### Prefrontal-Hippocampal Implants

After asymptotic training, all rats were implanted with chronic silicon probes. Five rats were implanted with 32-channel silicon probes arranged as tetrodes in the prefrontal cortex (layer VI, Supplementary Fig. 1A$_i$) and hippocampus (slm, Supplementary Fig. 1A$_{ii}$, B$_{ii}$) with 25 μm between adjacent electrode sites and impedances of $1.23 \pm 0.32$ MΩ. The 8 tetrodes were distributed across 4 shanks at tip-to-tetrode depths of 78 μm and 228 μm (NeuroNexus A4X2-tet-5 mm). Shanks were separated by 200 μm, giving each probe a total length of 0.67 cm. During implantation, the long axis was oriented medial-lateral. One rat was implanted with a silicon probe targeting prefrontal cortex (A/P 3.24 mm, M/L 0.6–1.2 mm, D/V$_{\text{from cortex}}$ −3.45 mm). Four rats were implanted with silicon probes targeting both prefrontal cortex (A/P 3.24 mm, M/L 0.6–1.2 mm, D/V$_{\text{from cortex}}$ −3.45 mm) and hippocampus (specifically CA1; A/P −3.24 mm, M/L 2.4–3.0 mm, D/V$_{\text{from cortex}}$ −2.4 mm). An additional 4 rats were implanted with 32-channel silicon probes in the RE and hippocampus with impedances of 0.043 MΩ (Supplementary Fig. 1B$_i$). The 32 single electrodes were distributed across 2 shanks spanning a length of 300 μm from the tip (Cambridge NeuroTech AASY-116 E-1 & E-2). Shanks were separated by 250 μm. During implantation, the long axis was oriented medial-lateral. Four rats were implanted with silicon probes targeting both the RE (A/P 1.92 mm, M/L 0.3–0.55 mm, D/V$_{\text{from cortex}}$ 7.0 mm) and hippocampus (specifically CA1; A/P 3.24 mm, M/L 2.4–2.65 mm, D/V$_{\text{from cortex}}$ 2.5 mm).

**Prefrontal-Hippocampal and RE-Hippocampal surgery.** Rats were anesthetized with isoflurane (induction 5%; maintenance: 2% to 3%) mixed with oxygen (800 ml/min) and placed in a stereotaxic apparatus (David Kopf Instruments, Model 900). A protective ophthalmic ointment (Gentak, 0.3%) was applied to the eyes, and the scalp was sterilized with applications of isopropyl alcohol (70% in deionized H$_2$O) followed by Betadine. The incision site was locally anesthetized with Marcaine® (7.5 mg/ml, 0.5 ml, s.c.) and the skull was exposed following a fisheye incision. Adjustments were made to ensure that bregma and lambda were level ( ± 0.05 μm in the D/V plane). Body temperature (35.9–37.5 °C) was monitored and maintained throughout surgery using a rectal thermometer and circulating water heating pad. Ringer's solution with 5% dextrose was administered to maintain hydration (5 ml, s.c.), and glycopyrrolate (0.2 mg/ml, 0.5 mg/kg, s.c.) was administered to prevent respiratory difficulties.

For silicon probe implants using the RatHat[64] targeting prefrontal cortex (*n* = 1), a rectangular craniotomy was drilled to accommodate the probe shanks centered on coordinates AP 3.24 mm, ML 0.9 mm. For dual-site implants in the prefrontal cortex and hippocampus (*n* = 4), 2 craniotomies were drilled targeting the prefrontal cortex (AP 3.24 mm, ML 0.9 mm) and hippocampus (AP − 3.24 mm, ML 2.7 mm). For dual-site implants using the RatHat[64] targeting the RE and hippocampus (*n* = 4), 2 craniotomies were drilled targeting the RE (AP 1.92 mm, ML 0.3–0.55 mm, DV$_{\text{from cortex}}$ 7.0 mm) and hippocampus (specifically CA1; AP 3.24 mm, ML 2.4–2.65 mm, DV$_{\text{from cortex}}$ 2.5 mm). Burr holes were drilled and skull screws (1/8-inch grade 2 (CP) titanium; Allied Titanium Inc) were secured onto the skull. After removal of the dura in the craniotomy, the implants were lowered via the stereotaxic arm until the electrode tips were just above the cortical surface. The ground wire was attached to the ground screws, and the implants were lowered such that prefrontal cortex electrodes reached ~3.3 mm below the cortical surface and hippocampal electrodes reached ~2.4 mm below the cortical surface. We first applied a 0.5% sodium alginate solution using a syringe with a 23 G needle to the exposed brain tissue within the craniotomy, followed by several drops of a 10% calcium

chloride solution to fix the alginate into a gel. The silicon probe was affixed to the surgical screws with dental cement (methyl, methacrylate). The RatHat[64] was created and assembled around the exposed silicon probe to protect the headstage against impact and debris. Excess skin was sutured (black silk suture 4-0, with reverse cutting needle 19 mm, 1/2 Circle). Neosporin® was applied to the skin surrounding the head stage. At the end of surgery, flunixin (50 mg/ml, 2.5 mg/kg, s.c.), a nonsteroidal anti-inflammatory analgesic, was administered to the rats. The rats were returned to a clean cage and monitored until they awoke. One day following surgery, the headstage was checked, the rats were administered a dose of flunixin, and Neosporin® was reapplied. At least 1 week was allowed for recovery from surgery prior to beginning experiments. All rats were considered for analysis.

## Prefrontal-Hippocampal and RE-Hippocampal Electrophysiological Recordings

Rats were recorded during the sequence memory task for 3 consecutive sessions. Throughout each experiment, wide-band data was acquired with digital headstages (32-channel, 40 kHz sampling rate) and OmniplexD systems (256-channel, Plexon) coupled to an automated behavioral rig. The OmniplexD systems acquires 32 digital behavioral event inputs, 16 analog event inputs, and digital video (80 fps). Voltage signals recorded from the tetrode tips were referenced to a ground screw positioned over the cerebellum (low cutoff=0.7 Hz). LFPs were separated into a second datastream (LFP: 0.7–300 Hz). LFP frequency bands of interest, including delta (1–4 Hz), theta (6–12 Hz) and beta (15–30 Hz), were filtered offline. LFP activity was analyzed with NeuroExplorer (Plexon) and MATLAB 2021a.

## Optogenetic experiment

**RE surgery.** Rats ($n = 10$) were acclimated to the housing facility and handled for at least 1 week prior to the initiation of any surgical procedures. In preparation for surgery, rats were anesthetized using isoflurane and their heads fixed in a stereotaxic apparatus (Kopf). An incision was made on the skin to expose the skull and burr holes were drilled above the prelimbic/infralimbic cortex (AP: +3.0, ML: ±0.8); midline thalamus (AP: −2.0, ML: −0.2), and ventral hippocampus (AP: −5.6, ML: ±5.85) using a RatHat surgical stencil[64].

Bilateral injections of channelrhodopsin-infused retrograde virus (retro pAAV-Syn-ChR2(H134R)-GFP; $n = 5$, 3 females) or retrograde control virus (retro AAV-CAG-GFP; $n = 5$, 2 females) were delivered to the ventral hippocampus at the following coordinates (AP: −5.6, ML: ±5.85; DV: −7.4, 200 nL, −6.8, 200 nL & −6.2, 100 nL) at 1.0 nL/s (Nanoject III, Drummund Scientific).

After the injections, the needle was kept in place for an additional 5 min to facilitate diffusion. Negative pressure was then used to avoid upward suction of the virus during tip retraction from the brain. Upon delivering the viral vector(s), a custom-built microelectrode-optrode RatHat[64] assembly was implanted. The RatHat contained stainless steel wire electrodes that recorded LFP (wire length from skull: prelimbic/infralimbic 5.5–6 mm; and ventral hippocampus 5.5–6 mm). At the conclusion of the surgery, the implants were secured to the skull using titanium bone screws and dental cement. Animals recovered for 3–4 weeks to allow for viral vector expression prior to starting the electrophysiological recordings. Microelectrode impedances were tested at 4 intervals (prior to implantation, 1 week after surgery, 1 day before testing began, and prior to perfusion) to check out-of-range impedances and dead channels. Rats with incorrect placement of the optic fiber or stainless-steel wires, improper injection site or viral expression, or defective electrodes ($n = 1$) were not included in the analysis.

## RE Optic Fiber Placement

At 5 days before beginning the recording sessions, rats were lightly anesthetized for chronic placement of an in-house fabricated optic fiber stub (tip Ø: 200 μm, 0.5 NA; Thorlabs). The length of the fiber stub was customized to deliver light intracranially above the RE (AP:2.0, ML: +0.2; DV length 7.0 mm) via a cannula in the RatHat implant. Then the optic fiber was secured with dental cement. Finally, optrode fiber light delivery was calculated and documented prior to and after each session to ensure light delivery remained stable (10–12 mW) throughout the experiment using a power meter (Thorlabs).

## Open field behavioral apparatus

Open-field (OF) recordings occurred in a rectangular maze (OF; dimensions: 122 cm × 118 cm x 47 cm) placed 72 cm above the floor (described previously[34]). The OF was surrounded by black curtains and room lights were kept off except for red LED lights above the arena.

## Optical Stimulation

Virally transfected cells (with and without ChR2) were optically stimulated using a 465 nm (blue light) compact LED attached to a rotary commutator above the OF (Plexon). The LED was connected to the optic fiber stub implanted in the brain via a patch cable (1.75 m long). Photostimulation events were driven with a PlexBright 4 channel controller and Radiant V2.3.0 software (Plexon). Animals received pulse and sine wave stimulations according to their recording schedule (see below). RE neurons were optically stimulated with blue light (465 nm) in 5 min blocks (min, ON-blocks) using 60 ms ON-pulses at 1-, 2-, 4-, or 8 Hz frequencies, and 20 ms ON-pulses at 50 Hz across 3 recording sessions (ascending, shuffled, and descending frequency order). On separate days, continuous sinewave stimulations were delivered in 5 min blocks at 1-, 2-, 4-, 8-, or 20 Hz frequencies. In all recording sessions, the first 5 min of recording (baseline) and the 5 min periods after each stimulation block were not optically stimulated. These OFF-blocks prevented overheating of the brain tissue and allowed activated neurons to return to baseline cell activity before the start of the next stimulation block.

## RE Optogenetic Recordings

At 7 days before the first recording session, rats were habituated to the OF arena and recording equipment. During habituation days, rats were allowed to freely explore the OF for 20 min and restrained lightly to connect them to the tethered and optic patch cable. On the last 2 days of habituation, the rats were familiarized to food treats (each treat: ~1/8 fruit Loops) that were thrown into the maze from above every 5 min to encourage rats to move around the arena. Neural data was acquired with a Plexon OmniplexD recording system (wide band: 40 kHz) and the PlexControl data acquisition software via implanted 50 μm Teflon-coated stainless steel electrode wires connected to an electrode interface board (Neuralynx). LFP were sampled at 1 kHz.

## Histology

After completing all experiments, the rats were anesthetized with isoflurane (5%) mixed with oxygen (800 ml/min), and marking lesions were made with a NanoZ (Plexon) to deliver 12 A for 10 s to each of the electrode locations. Rats were then transcardially perfused with 100 ml phosphate-buffered saline (PBS), followed by 200 ml of 4% paraformaldehyde (pH 7.4; MilliporeSigma). Brains were post-fixed overnight in 4% paraformaldehyde and placed in a 30% sucrose solution for cryoprotection. Frozen brains were cut on a Leica CM3050 S cryostat (40 μm; coronal plane) into 3 sets of immediately adjacent sections. One set was mounted, Nissl-stained, and coverslipped with Permount to visualize marking lesions. Marking lesions were then mapped onto plates from Paxino and Watson[65]. Marking lesions were then mapped onto plates from Paxino and Watson[65].

In optogenetic subjects, an additional set of sections was incubated at room temperature for 48 h with anti-green fluorescent protein (GFP) rabbit primary antibody (1:500, Rockland). After washing,

the tissue was incubated for 6 h at room temperature in VectaFlour DyLight 488 anti-rabbit secondary antibody (1:500, Vector Labs). After incubation, tissue was washed in 0.1 M PB (3 × 5 min) then mounted on gelatin-coated slides and coverslipped with VectaShield mounting medium.

## Quantification and statistical analysis

**Sequence memory performance analysis.** Performance on the task can be analyzed using a number of measures[63,66]. The first position of each sequence was excluded from all analyses as these items are always InSeq. Expected vs. observed frequencies were analyzed with G tests to determine whether the observed frequencies of InSeq and OutSeq responses for a given session were significantly different from the frequency expected by chance. G tests provide a measure of performance that controls for response bias and is a robust alternative to the $\chi^2$ test, especially for datasets that include cells with smaller frequencies[67]. To compare performance across sessions or animals, we calculated the SMI[3,36,63].

In essence, the SMI normalizes the proportion of InSeq and Out-Seq items presented during a session and reduces sequence memory performance to a single value ranging from −1 to 1. A score of 1 represents perfect sequence memory, a score of 0 indicates chance performance, and a negative SMI score represents performance below chance levels.

## LFP analyses

Custom scripts using core MATLAB 2021a functions were used with native signal processing functions, the Chronux toolbox (http://chronux.org/)[68], and circ_stats toolboxes (Linearize_Circle) for all analysis. We plotted spectrograms for each session and aligned them to various timestamps (poke-out for sequence task and stimulation events for optogenetic blocks) to observe the different frequencies during relevant events. Perievent spectrograms and coherograms were used to visualize and quantify LFP activity using custom scripts using MATLAB 2021a functions (imagesc, spectrogram, contour). In the optogenetic experiment, we quantified and averaged 1 s periods of raw voltage signals relative to the onset of the 8-Hz blue light pulses across days and 4-, 8-, and 20 Hz sinewaves to determine if stimulation evoked responses in downstream regions and compare the efficacy of our stimulation patterns (Fig. 4C, Supplementary Fig 8). We calculated coherence using 'mscohere' and plotted across frequencies in 0.4883 Hz steps. We focused our analyses on the 1–4 Hz (delta), 6–12 Hz (theta), and 15–30 Hz (beta), as our spectrograms showed high power in those bands at the level of individual rats and groups. We calculated the AUC using trapz (trapezoidal numerical integration) to measure coherence magnitudes based on typical magnitude-squared coherence calculations (mscohere) across the entire frequency range for predefined bands of interest (e.g., Theta: 6–12 Hz or Beta: 15–30 Hz; see[69,70]). We compared across animals by z-normalizing values to calculate the t-score in spectrograms and AUC for coherence (zscore). We used traditional (parametric) t tests and ANOVAs for behavioral and frequency-based statistical comparisons (mean, std, ttest). Tests were considered significant at $p < 0.05$.

## Behavioral Tracking

Behaviorally relevant events were identified through behavioral tracking using DeepLabCut (DLC)[71] for sequence memory experiments. Markers were placed manually on the rat's nose, left and right ears, center of body/mass, tail base, and mid-tail. Markers were tagged on 200–350 (per experiment) frames extracted from the recording video sessions to create a training dataset and used to train the network using iterations. After the initial training, output video files containing the continued markers and skeleton connecting structure were reviewed to identify out of place markers. Sample markers that had likelihood below 0.90 were refined until they met criteria.

The exploratory behaviors (running and stationary periods) of rats in the sequence memory linear track were identified from the 'x' and 'y' markers' coordinates per frame (DLC output) using MATLAB 2021a. These bouts were obtained using previously defined exclusion criteria[34,72–74]. For running bouts, the minimum duration was specified as 2.05 s with speeds above 2.5 cm/s (achieved using a median filter). For stationary bouts, the duration was also at least 2.05 s with speeds under 2.5 cm/s (limited for linear track). Behaviors such as grooming and rearing were excluded according to the exclusion criteria.

## Beta burst detection

An envelope analysis was performed with the raw LFP data to obtain z-scored upper envelopes (curves for the boundaries of the waveform) for all trials, which were averaged for each session. The raw data and z-scored envelopes using 'envelope' were visually inspected for beta rhythm bursts across trials. A set of exclusionary parameters was defined based on this inspection to analyze beta bursts that met clear bursting criteria. Bursts were defined by minimum onset and offset amplitude values of 0.15 in the z-scored envelope, with a minimum peak in amplitude of 1.30 between the start and end of a burst. These were further separated from bursts that lasted under 0.95 ms so that bursts with at least 2 complete cycles were considered in the raw waveform. We analyzed InSeq_correct trials to calculate burst latencies which refers to the time that elapsed between the start of the trial and the beginning of the first beta burst. The mean time of the beta burst latency across trials, sessions, and subjects was calculated, as well as mean duration of the bursts in each region. Similarly, a cumulative density function was used to compare the beta onsets between regions (co-occurrences of bursts).

## Reporting summary

Further information on research design is available in the Nature Portfolio Reporting Summary linked to this article.

## Data availability

All data supporting the findings in this paper, and its supplementary figures, are available from the corresponding author upon request. Source data for figures are provided with this paper. Source data are provided with this paper.

## Code availability

All the scripts used in the current study were written and executed around built-in functions in MATLAB 2021a and are available from the corresponding authors upon reasonable request.

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

## Acknowledgements
This work was supported by National Institutes of Health (NIH) grants R01 MH113626 (T.A.A) and F99 NS119001 (M.J), FIU and the Feinberg Foundation. A special thanks to all the members of the Allen Lab, specifically our undergraduate research assistants, F.P.R., N.T., G.E.R., D.S., N.L., C.R., K.K.R., and J.M., who helped with task training and data collection, pre-processing, and histology. We would like to thank the Animal Care Facility and Dr. Horatiu Vinerean.

## Author contributions
Conceptualization, M.J., T.D.V., and T.A.A.; Methodology, M.J., T.D.V., and T.A.A.; Investigation, M.J. and T.D.V., A.G., A.V.V, S.L, and V.R; Writing–Original Draft, M.J., T.D.V., and T.A.A.; Writing–Review and Editing, T.A.A., M.J., T.D.V., A.G., A.V.V., S.L., V.R., and R.P.V.; Funding Acquisition, T.A.A., R.P.V., T.D.V., and M.J.; Resources, T.A.A., R.P.V.; Supervision, T.A.A.

## Competing interests
The authors declare no competing interests.
