## [Peer Review File · Nature Communications]

Nucleus reuniens transiently synchronizes memory networks at beta frequenciesREVIEWER COMMENTS

Reviewer #1 (Remarks to the Author):

The reuniens nucleus of the thalamus is thought to organize oscillatory synchrony between the hippocampus and the prefrontal cortex, hence playing a major role in memory. Previous works have shown the reuniens nucleus involvement in slow oscillation, delta, hippocampal theta and cortical gamma. The authors of this manuscript investigate the role of the reuniens nucleus in beta oscillations (15-30Hz) and its importance in flexible memory and sensory integration.

The authors cleverly combined pairs of local field potential recordings (prefrontal cortex-hippocampus or reuniens-hippocampus) to a non-spatial sequence memory task in rats. This approach allowed to study the oscillation patterns in the different brain regions during a memory task that is not contaminated by brain activity related to running. The analysis reported are appropriate and the experiments are well controlled.

Interestingly, the authors observed burst of beta activity in all three regions that closely match odor sampling. Beta activity rose earlier and showed stronger amplitude in the reuniens nucleus and was quickly followed by a rise of beta activity in both mPFC and hippocampus. Overall, the coherence in beta rhythm was increased between these three brain regions during the sequence memory task. Optogenetic activation of the reuniens nucleus was sufficient to produce a similar increase in coherence between mPFC and hippocampus in freely behaving rats. The presentation of the results is clear and accessible and the authors provided complete extended data figures that are helpful to the reader.

Overall, I consider this manuscript to be a very good paper that contributes significantly to our understanding of oscillations synchrony between the mPFC and hippocampus and provides further evidence on the distinct role of beta activity in memory network.

I do not have major comments.

Minor comments:

- Discussion, lines 334-336. This manuscript is convincingly showing that (1) beta activity in mPFC-reuniens-hippocampus circuit accounted for a majority of the coherence in the network during the sequence memory task and (2) reuniens can drive prefrontal-hippocampal beta coherence in freely behaving rats. It also strongly suggests that the reuniens nucleus could drive the beta coherence in the mPFC-Re-hippocampus network during the sequence memory task. However, to demonstrate this causal link, perturbing the reuniens activity during the sequence memory task would be necessary (e.g. optogenetic inhibition of reuniens during odor sampling could alter beta burst and performance). This challenging experiment might be beyond the scope of this paper. I would strongly suggest to rephrase the first sentence of the discussion to reflect that the paper in this current form do not demonstrate a causal link between reuniens induced beta coherence in mPFC-hippocampus and sequence memory task-induced beta coherence in the same network.

- Legend Fig2G: typo "The ..."

- Fig4E: It seems that there is a significant increase in mPFC-HC coherence in delta frequencies during and after light stimulation in ChR2+ rats that is not discussed in the text.

- Extended data Fig8: are state transitions induced by sine stimulation at low frequencies specific to sine stimulation? Can they be triggered by square pulses with a similar duration?

Reviewer #2 (Remarks to the Author):

In this manuscript, the authors measure neural synchrony between three brain areas (the medial prefrontal cortex, dorsal hippocampus, and nucleus reuniens of the thalamus) during a sequence memory task in male and female rats. Inter-area neural synchrony is assessed via phase and amplitude coherence between local field potentials. The authors also measure amplitude and phase-amplitude coupling in each of these areas. The authors find that both power and coherence in the beta frequency band are increased in all brain areas during “memory-related” portions of the task (times at which the rat samples in-sequence and out-of-sequence odors via nose port), relative to stationary and non-stationary (running) “non-memory” portions of the task. The authors further find that beta power and coherence are increased specifically during correct “in-sequence” trials (during which the animal initiates a nosepoke that is >1 second in duration), as compared to incorrect “in-sequence” and both correct and incorrect “out-of-sequence” trials. Visual inspection and subsequent analysis of the in-trial timing of beta coherence revealed that “beta bursts” first occur in the nucleus reuniens, and subsequently emerge in the prefrontal cortex and hippocampus. The authors use these results as support for the hypothesis that the nucleus reuniens drives beta synchrony between the three areas during successful sequence memory.

As a step toward testing this hypothesis, the authors use optogenetics to stimulate neurons in the nucleus reuniens that send projections to the ventral hippocampus. The authors find that optogenetic stimulation of these projections suppresses theta coherence between the prefrontal cortex and hippocampus, and increases beta coherence between the two structures.

The overwhelming majority of research in the area of rodent memory systems has focused on theta oscillations (and, to a much smaller degree, gamma oscillations), especially when that research concerns the hippocampus and prefrontal cortex. The fixation on theta is likely explained, at least in part, by the fact that it dominates the hippocampal local field potential during active behavior (not as much in the prefrontal cortex, where lower-frequency delta oscillations are usually most dominant). As the authors point out, however, this is most probably a reflection of the structure of most rodent memory tasks, which usually require ambulation during the “memory-related” portions of each trial. Therefore, the finding that it is beta, rather than theta, that is increased during stationary “memory-related” trial epochs is quite novel and exciting, and these results will no doubt prove to be impactful and beneficial for memory researchers.

My broad evaluation of the manuscript is that it is well-written, the experimental methods are (mostly) well-described (see below), the experimental design makes sense, the authors' conclusions are in line with their results, and the results are convincing. I have some broad questions about aspects of the manuscript – these questions are outlined below. I believe that elaborating on these questions will improve understanding of the manuscript.

Questions:

For the optogenetic experiments – the use of a non-retro AAV with a CAG promoter as a control seems like an odd choice, given that the experimental virus was AAVretro with a synapsin promoter. In an ideal world, serotype and promoter would be matched between experimental and control viruses in order to increase the likelihood that similar cell types are infected between conditions. What was the rationale behind choosing the control virus for these experiments?

Can the authors explain the use of an “area under the curve” metric for phase coherence? I am unfamiliar with this way of analyzing coherence, and I believe the rationale for choosing this method over more “typical” methods (e.g., calculating the correlation between power spectral densities and cross-spectral densities) would be beneficial for the reader.

From the Introduction: Do the authors have a citation for the assertion that “the RE innervates both glutamatergic and GABAergic target cells in the prefrontal cortex and CA1...”? Also, what is meant by “excitatory-inhibitory tone”?

When comparing beta coherence between “memory”, “non-memory stationary”, and “running” conditions, are correct and incorrect trials collapsed for “memory” conditions? Or were incorrect “in-sequence” and “out-of-sequence” trials excluded?

Can the authors explain what the latency is from when calculating “latency to first burst”? Is it latency from the start of the trial? Latency from the previous nosepoke? This should be made clearer.

The authors optogenetically manipulate reuniens cells that target the ventral hippocampus (which makes sense given the anatomical connections between the two areas), but record LFPs from the dorsal hippocampus. What is the authors' hypothesis regarding how RE-vHC stimulation might impact dorsal hippocampal-mPFC beta coherence? How might this relate to memory-specific behavior during the sequence task?

Nature Communications (NCOMMS-22-43826A-Z) - Response to Reviewers

Nucleus reuniens transiently synchronizes memory networks at beta frequencies

Maanasa Jayachandran, Tatiana D. Viena, Andy Garcia, Abdiel Vasallo Veliz, Sofia Leyva, Valentina Roldan, Robert P. Vertes, Timothy A. Allen

We thank Reviewer 1 and Reviewer 2 for their thoughtful comments and overall positive evaluations. We have addressed all the comments raised in a point-by-point fashion below and made corresponding changes in the revised manuscript (denoted in blue text). As requested, the updated manuscript now includes new analyses on the optogenetic-based reuniens induction of prefrontal-hippocampal delta coherence, additional methodological details (and corrections), and revised language in the introduction/discussion. We believe the revised manuscript has been significantly improved by the changes.

Reviewer #1 (Remarks to the Author):

The reuniens nucleus of the thalamus is thought to organize oscillatory synchrony between the hippocampus and the prefrontal cortex, hence playing a major role in memory. Previous works have shown the reuniens nucleus involvement in slow oscillation, delta, hippocampal theta and cortical gamma. The authors of this manuscript investigate the role of the reuniens nucleus in beta oscillations (15-30Hz) and its importance in flexible memory and sensory integration.

The authors cleverly combined pairs of local field potential recordings (prefrontal cortex-hippocampus or reuniens-hippocampus) to a non-spatial sequence memory task in rats. This approach allowed to study the oscillation patterns in the different brain regions during a memory task that is not contaminated by brain activity related to running. The analysis reported are appropriate and the experiments are well controlled.

Interestingly, the authors observed burst of beta activity in all three regions that closely match odor sampling. Beta activity rose earlier and showed stronger amplitude in the reuniens nucleus and was quickly followed by a rise of beta activity in both mPFC and hippocampus. Overall, the coherence in beta rhythm was increased between these three brain regions during the sequence memory task. Optogenetic activation of the reuniens nucleus was sufficient to produce a similar increase in coherence between mPFC and hippocampus in freely behaving rats. The presentation of the results is clear and accessible and the authors provided complete extended data figures that are helpful to the reader. Overall, I consider this manuscript to be a very good paper that contributes significantly to our understanding of oscillations synchrony between the mPFC and hippocampus and provides further evidence on the distinct role of beta activity in memory network.

Major comments:

I do not have major comments.

Minor comments:

1. Discussion, lines 334-336. This manuscript is convincingly showing that (1) beta activity in mPFC-reuniens-hippocampus circuit accounted for a majority of the coherence in the network during the sequence memory task and (2) reuniens can drive prefrontal-hippocampal beta coherence in freely behaving rats. It also strongly suggests that the reuniens nucleus could drive the beta coherence in the mPFC-Re-hippocampus network during the sequence memory task. However, to demonstrate this causal link, perturbing the reuniens activity during the sequence memory task would be necessary (e.g. optogenetic inhibition of reuniens during odor sampling could alter beta burst and performance). This challenging experiment might be beyond the scope of this paper. *I would strongly suggest to rephrase the first sentence of the discussion to reflect that the paper in this current form do not demonstrate a causal*

link between reuniens induced beta coherence in mPFC-hippocampus and sequence memory task-induced beta coherence in the same network.

We agree and have rephrased the first sentence on **line 343-345** of the Discussion as the Reviewer suggests for clarity. It now reads: “Our findings revealed that 15–30 Hz beta activity accounted for a majority of the coherence in prefrontal-reuniens-hippocampal interactions during sequence memory retrieval, and that reuniens neurons can drive 15-30 Hz beta coherence between the prefrontal cortex and hippocampus in freely-behaving rats.”

2. Legend Fig2G: typo “The ...”

We corrected this on **line 555**.

3. Fig4E: It seems that there a significant increase mPFC-HC coherence in delta frequencies during and after light stimulation in Chr2+ rats that is not discussed in the text.

We examined coherence, *a priori*, in the theta and beta bands and so reported on these findings. However, as the Reviewer noticed, we also observed notable prefrontal-hippocampal delta coherence following optogenetic stimulation of reuniens neurons. We have analyzed the delta band coherence and now report the results alongside the beta and theta results in the text on **lines 317-324** and in **Figure 4**. These findings further support the notion of delta-beta relationships.

4. Extended data Fig8: are triggered state transitions induced by sine stimulation at low frequencies specific to sine stimulation? Can they be triggered by square pulses with a similar duration?

State transitions appear to occur most reliably near the rising and falling phase of the sine stimulations (2,4,8, and 20Hz), which is not the case with the onset and offset of the pulse stimuli. While column B shows rapid potential deflections around ~0V, column C shows linear shifts to a steadier elevated or lowered positive or negative voltage levels. Based on these data, we think ramping stimuli (saw tooth; rising or falling) may be better for inducing rapid state transitions reflecting the rising and falling phases of the sine stimulations. Notably though, the pulse stimuli are clearly sufficient to induce overall changes in the spectral modes within the target regions. We discuss this in the Extended Data Figure 8 captions in **lines 952-955**.

Reviewer #2 (Remarks to the Author):

In this manuscript, the authors measure neural synchrony between three brain areas (the medial prefrontal cortex, dorsal hippocampus, and nucleus reuniens of the thalamus) during a sequence memory task in male and female rats. Inter-area neural synchrony is assessed via phase and amplitude coherence between local field potentials. The authors also measure amplitude and phase-amplitude coupling in each of these areas. The authors find that both power and coherence in the beta frequency band are increased in all brain areas during “memory-related” portions of the task (times at which the rat samples in-sequence and out-of-sequence odors via nose port), relative to stationary and non-stationary (running) “non-memory” portions of the task. The authors further find that beta power and coherence are increased specifically during correct “in-sequence” trials (during which the animal initiates a nosepoke that is >1 second in duration), as compared to incorrect “in-sequence” and both correct and incorrect “out-of-sequence” trials. Visual inspection and subsequent analysis of the in-trial timing of beta coherence revealed that “beta bursts” first occur in the nucleus reuniens, and subsequently emerge in the prefrontal cortex and hippocampus. The authors use these results as support for the hypothesis that the nucleus reuniens drives beta synchrony between the three areas during successful sequence memory.

As a step toward testing this hypothesis, the authors use optogenetics to stimulate neurons in the reuniens that send projections to the ventral hippocampus. The authors find that optogenetic stimulation of these projections suppresses theta coherence between the prefrontal cortex and hippocampus, and increases beta coherence between the two structures.

The overwhelming majority of research in the area of rodent memory systems has focused on theta oscillations (and, to a much smaller degree, gamma oscillations), especially when that research concerns the hippocampus and prefrontal cortex. The fixation on theta is likely explained, at least in part, by the fact that it dominates the hippocampal local field potential during active behavior (not as much in the prefrontal cortex, where lower-frequency delta oscillations are usually most dominant). As the authors point out, however, this is most probably a reflection of the structure of most rodent memory tasks, which usually require ambulation during the “memory-related” portions of each trial. Therefore, the finding that it is beta, rather than theta, that is increased during stationary “memory-related” trial epochs is quite novel and exciting, and these results will no doubt prove to be impactful and beneficial for memory researchers.

My broad evaluation of the manuscript is that it is well-written, the experimental methods are (mostly) well-described (see below), the experimental design makes sense, the authors’ conclusions are in line with their results, and the results are convincing. I have some broad questions about aspects of the manuscript – these questions are outlined below. I believe that elaborating on these questions will improve understanding of the manuscript.

Questions:

1. For the optogenetic experiments – the use of a non-retro AAV with a CAG promoter as a control seems like an odd choice, given that the experimental virus was AAVretro with a synapsin promoter. In an ideal world, serotype and promoter would be matched between experimental and control viruses in order to increase the likelihood that similar cell types are infected between conditions. What was the rationale behind choosing the control virus for these experiments?

We clarify that we did use a retrograde-AAV with our control rats (Addgene 37825-AAVrg). We now include the word ‘retrograde’ before AAV-CAG-GFP in Figure 4 (line 598) and line 711 of the manuscript to make it clear to the reader we utilized a retrograde virus on both conditions.

We agree with the reviewer that the selection of a proper promoter is important in any experimental design. While there are differences in the type of cells hSyn and CAG can target, due to the nature of our retrograde technique (injections in ventral hippocampus for the retrograde labeling of neurons in reuniens); previous findings in our lab show that this specific retro-AAV-CAG retrograde vector labels distinct populations of projecting neurons in reuniens (see Viena et al., 2020); and based on the anatomy of the circuit, we feel confident that the targeted reuniens neurons are projecting reuniens-to-ventral hippocampal neurons, leaving out the possibility of retrogradely labeling interneurons within reuniens.

2. Can the authors explain the use of an “area under the curve” metric for phase coherence? I am unfamiliar with this way of analyzing coherence, and I believe the rationale for choosing this method over more “typical” methods (e.g., calculating the correlation between power spectral densities and cross-spectral densities) would be beneficial for the reader.

Area under the curve (AUC) was used to measure coherence magnitudes based on typical magnitude-squared coherence calculations (MATLAB ‘mscohere’ function) across the entire frequency range for predefined bands of interest (e.g., Theta: 6-12 Hz or Beta: 15-30Hz) using trapezoidal numerical integration (MATLAB ‘trapz’ function). Similar applications have been used in other spectral analysis, for example see Gwilt, Bauer, & Bast (2020) and Iskhakova et al. (2021). We clarify these important methodological details on lines 795-800.

3. From the Introduction: Do the authors have a citation for the assertion that “the RE innervates both

glutamatergic and GABAergic target cells in the prefrontal cortex and CA1...”? Also, what is meant by “excitatory-inhibitory tone”?

Several studies have demonstrated that reuniens sends excitatory glutamatergic projections to the slm layer of the CA1 region of the hippocampus and the medial prefrontal cortex (Dolleman-van der Weel et al., 1997, 2017; Dolleman-van der Weel and Witter 2000; Wouterlood et al. 1990; Colbert and Levy 1992; Bokor et al. 2002 Hur and Zaborsky 2005; Viana Di Prisco and Vertes, 2006; Eleore et al., 2011; Cruikshank et al., 2012; Goswamee, Legget & McQuiston, 2021 and Topolnik & Tamboli, 2022) synapsing on both glutamatergic and GABAergic cells (see Dolleman van der Weel, 2019 review; esp. Figure 2 and 4). Along these lines, Dolleman van der Weel (1997, 2017) showed that stimulation of reuniens in anesthetized rats leads to strong fEPSPs centered on the slm, but did not lead to spiking activity in the pyramidal layer presumably because the feed-forward inhibition overwhelms the excitatory inputs under these conditions. The same effect was obtained in medial prefrontal cortex (Di Prisco and Vertes, 2006; Cruikshank et al. 2012). More recently, these effects were demonstrated recently using *ex vivo* optogenetic slice physiology experiments demonstrating monosynaptic feed-forward excitation and polysynaptic feed-forward inhibition (Vantomme et al., 2022, Society for Neuroscience Poster 763.25, San Diego, CA). In this way, reuniens inputs will change the overall balance of excitatory and inhibitory influences on projection neurons in the CA1 and prefrontal target regions, which is what we meant by “excitatory-inhibitory tone”. We added appropriate references on **line 92**.

4. When comparing beta coherence between “memory”, “non-memory stationary”, and “running” conditions, are correct and incorrect trials collapsed for “memory” conditions? Or were incorrect “in-sequence” and “out-of-sequence” trials excluded?

For this analysis the “memory” condition refers to correct trials regardless of sequential context (i.e., Inseq Correct and Out-of-Sequence Correct trials only). A correct trial suggests memory retrieval whereas an incorrect trial is more ambiguous. We further clarify this in the results on **lines 101-103**.

5. Can the authors explain what the latency is from when calculating “latency to first burst”? Is it latency from the start of the trial? Latency from the previous nosepoke? This should be made clearer.

In this analysis latency refers to the time that elapses between the start of the trial and the beginning of the first beta burst. We clarified this on **lines 823-825** in the methods.

6. The authors optogenetically manipulate reuniens cells that target the ventral hippocampus (which makes sense given the anatomical connections between the two areas), but record LFPs from the dorsal hippocampus. What is the authors’ hypothesis regarding how RE-vHC stimulation might impact dorsal hippocampal-mPFC beta coherence? How might this relate to memory-specific behavior during the sequence task?

For the optogenetic experiments, we injected the retroAAV in ventral slm and recorded LFP’s from ventral slm for the reasons mentioned by the reviewer, notably that reuniens is reported to have 10 times more connections with the ventral hippocampus. Preliminarily, we have recorded from the dorsal hippocampus and stimulated reuniens optogenetically in a couple rats (n=2). We observed similar changes in the beta, theta and delta coherence bands. However due to the electrode placement, the preliminary low number of subjects, and recording differences we cannot say for sure what the effect would be physiologically. Speculatively, we think boosting reuniens activity during the odor stimulus will boost directed retrieval (Eichenbaum, 2017) via delta-beta rhythms, but possibly impair performance if stimulation occurs before or after the odor presentation which may be more critical for other theta-dominated memory functions (aka Figure 3; or for example, Shahbaba et al.,

2022; Hallock et al., 2022). We have expanded the discussion to address this comment on **lines 356-362**.

REVIEWERS' COMMENTS

Reviewer #1 (Remarks to the Author):

The authors satisfactorily addressed my concerns through changes to the manuscript and new analyses. The changes improved the manuscript and I have no further comments.

Reviewer #2 (Remarks to the Author):

I thank the authors for their responses to my original comments. I have nothing further to add.

Nature Communications (NCOMMS-22-43826A-Z) - Response to Reviewers

Nucleus reuniens transiently synchronizes memory networks at beta frequencies

Maanasa Jayachandran, Tatiana D. Viena, Andy Garcia, Abdiel Vasallo Veliz, Sofia Leyva, Valentina Roldan, Robert P. Vertes, Timothy A. Allen

We thank the reviewers for their thoughtful comments and time.

Reviewer #1 (Remarks to the Author):

The authors satisfactorily addressed my concerns through changes to the manuscript and new analyses. The changes improved the manuscript and I have no further comments.

Reviewer #2 (Remarks to the Author):

I thank the authors for their responses to my original comments. I have nothing further to add.